# Net ecosystem exchange and energy fluxes measured with eddy covariance technique in a West Siberian bog

Pavel Alekseychik[1], Ivan Mammarella[1], Dmitri Karpov[2], Sigrid Dengel[6], Irina Terentieva[3], Alexander Sabrekov[2,3], Mikhail Glagolev[2,3,4,5] and Elena Lapshina[2]

[1]Department of Physics, P.O. Box 68, FI-00014, University of Helsinki, Finland

[2]Department of Environmental dynamics and global climate change, 628012, Yugra State University, Russia

[3]Tomsk State University, Russia

[4]Department of Soil Physics and Development, 119991, Moscow State University, Russia

[5]Institute of Forest Science, Russian Academy of Sciences, 143030, Moscow, Russia

[6]Climate and Ecosystem Sciences Division, Lawrence Berkeley National Laboratory, Berkeley, USA

*Correspondence to*: Pavel Alekseychik (pavel.alekseychik@helsinki.fi)

**Abstract.** Very few studies of ecosystem-atmosphere exchange involving eddy-covariance data have been conducted in Siberia, with none in West Siberian middle taiga. This work provides the first estimates of carbon dioxide ($CO_2$) and energy budgets at a typical bog of the West Siberian middle taiga based on May-August measurements in 2015. The footprint of measured fluxes consisted of homogeneous mixture of tree-covered ridges and hollows with the vegetation represented by typical sedges and shrubs. Generally, the surface exchange rates resembled those of pine-covered bogs elsewhere. The surface energy balance closure approached 100%. Net $CO_2$ uptake was comparatively high, summing up to 202 gC m$^{-2}$ for the four measurement months, while the Bowen ratio was seasonally stable at 28%. The ecosystem turned into a net $CO_2$ source during several front passage events in June and July. Several periods of heavy rain helped keep the water table at a sustainably high level, preventing a usual drawdown in summer. However, because of the cloudy and rainy weather, the observed fluxes might rather represent the special weather conditions of 2015 than their typical magnitudes.

## 1. Introduction

Boreal peatlands, covering a large fraction of the northern hemisphere, are an important terrestrial carbon pool, whose size is estimated to be around $500 \pm 100$ Pg of organic carbon when integrated over the entire peat depth (Yu, 2012). Photosynthesis and respiration of plant and microbial communities regulate the size of this pool. However, peatlands are also prone to rapid ecological changes related to climate, which modify the interaction between hydrology, carbon cycle, vegetation cover and micro-topography. Detailed knowledge of the processes governing the carbon exchange in northern peatlands over the course of a growing season is limited, especially with respect to the impact of relevant environmental variables.

While continuous and long-term time series of carbon dioxide ($CO_2$), sensible and latent heat fluxes are already available from several boreal peatland sites in Europe, measurements of this kind are rare in Siberia. The closest permanent West-Siberian bog installation is part of the ZOTTO facility (Heimann et al. 2014), while other comparable stations are found in European Russia (Ust Pojeg - Gazovic et al. 2010), Finland (Tervalamminsuo – Annalea Lohila, personal communication; Siikaneva bog site - Korrensalo et al. 2017), Sweden (Fäjemyr - Lund et al. 2007), and Canada (Mer Bleue - Humphreys et al. 2014). This is mainly due to the lack of developed measurement sites with the infrastructure suitable for continuous monitoring of the ecosystem-atmosphere exchange processes, and general inaccessibility of key ecological zones and biomes. In remote and large areas such as West Siberia, current estimates of greenhouse gas exchange rates are largely uncertain, because discontinuous and short-term observations (static chamber technique) have often been used to derive regional and long term exchange rates (Golovatskaya and Dyukarev, 2008;Schneider et al., 2011;Glagolev et al., 2011;Sabrekov et al., 2013). Currently, only about ten eddy-covariance (EC) flux tower sites are active in Russia east of the Urals (Alekseychik et al., 2016). No prior publications of eddy-covariance fluxes from the West-Siberian peatlands are known to the authors. Previous studies utilising the EC method in Boreal peatlands elsewhere have shown the importance of temperature, solar radiation, and water table depth in controlling the net ecosystem exchange (NEE) (Arneth et al., 2002;Aurela, 2004;Lafleur et al., 2003;Friborg et al., 2003;Humphreys et al., 2006). Most studies show that, during the growing season, peatlands typically act as net sinks of $CO_2$. However, during warm and dry growing seasons the peatland sink strength is notably reduced and in some cases lead to net $CO_2$ losses (Bubier et al., 2003;Lafleur et al., 2003).

In order to fill the West Siberian measurement gap, we have recently established a new EC flux tower at the raised bog site at the Mukhrino field station in Khanty–Mansi Autonomous Okrug (Russia). The Mukhrino field station is officially part of the PEEX station network (Alekseychik et al., 2016) and INTERACT (http://www.eu-interact.org/). The energy and carbon dioxide flux data provided by the eddy-covariance tower is so far unique for West Siberian middle taiga and is the only setup functioning as of 2016 in West Siberia (within at least a radius of 1000 km). The aims of this study are to present and analyze the new flux data tower with the related ancillary measurements, to investigate the diurnal and seasonal variations of NEE and energy fluxes, as well as to determine summertime budgets of energy and $CO_2$ of the ecosystem.

## 2. Materials and methods

### 2.1 Site description

The Mukhrino Field Station (60°54' N, 68°42' E) is located at the eastern terrace of the Irtysh River 20 km south of the point of confluence with the Ob' river, in the middle taiga zone of the West Siberian Lowland. West Siberian Lowland is a geographical region of Russia bordered by the Ural Mountains in the west and the Yenisey River in the east; the region covers $2.75 \times 10^6$ km$^2$ from 62-89° E to 53-73° N. Paludification in West Siberian Lowland started after a climate warming 11500 cal. BP, with 55% of the present C store accumulated by 6 000 cal. BP. The mires have expanded very little during the past 3000 years (Turunen et al., 2001). The middle taiga ecozone (59-62° N) covers an area of about $0.57 \times 10^6$ km$^2$ in the central part of the West Siberian Lowland; the region features flat terrain with elevations of 80 to 100 m above sea level.

The region has a subarctic or boreal climate (Köppen-Geiger code Dfc) with long cold winter, short warm summer and frequent change of weather conditions. Average monthly air temperatures range from -20 to 18 °C over the year with the mean annual temperature of -1.1 °C. The latter increased by 0.4 °C from 1893-1935 to 1970-1999 period (Bulatov,

2007). Median annual precipitation is 520 mm, and evapotranspiration is 445 mm (Bulatov, 2007). Mean summer precipitation is 208 mm, ranging from 74 mm to 354 mm over the period from 1934 to 2014. Permafrost in any form is absent. Peat soils freeze to a depth of about 50 cm. Typically, snowmelt and river break up start in the first half of May. Mean duration of snow cover period is 180 days (from 19 October to 25 April) with the average March snow depth 54 cm. Growing season lasts for 98 days (Bulatov, 2007); the number of growing degree-days >5 °C is from 900 to 1500 (median – 1250).

The excess water supply and flat terrain with poor drainage provides favorable conditions for wetland formation in the region. Large wetland systems commonly cover watersheds (34% of the zonal area) and have a convex dome with centers that are 3 to 6 m higher than the periphery. The wetland subtypes here have strict spatial regularities. Ridge-hollow-lake complexes (15% of the total wetland area) represent central plateau depressions with stagnant water. They consist of numerous small lakes up to 2 m deep with peat bottom and waterlogged hollows. Different types of ridge-hollow complexes dominate (42%), covering the better drained gentle slopes. Pine bogs (28%) are more frequent in drier areas, where the peat surface is typically 10-50 cm above the water table level. Poor and rich fens (8%) develop along the wetland edges and watercourses, where the nutrient availability is higher. Open bogs with mosaic dwarf shrubs-sphagnum vegetation are widespread (5%) at the periphery of individual wetland bodies. Wooded swamps (2%) surround the peatland systems (Terentieva et al., 2015). Primary lakes of 100-2000 m in diameter and up to 5 meters depth with mineral bottom are widespread.

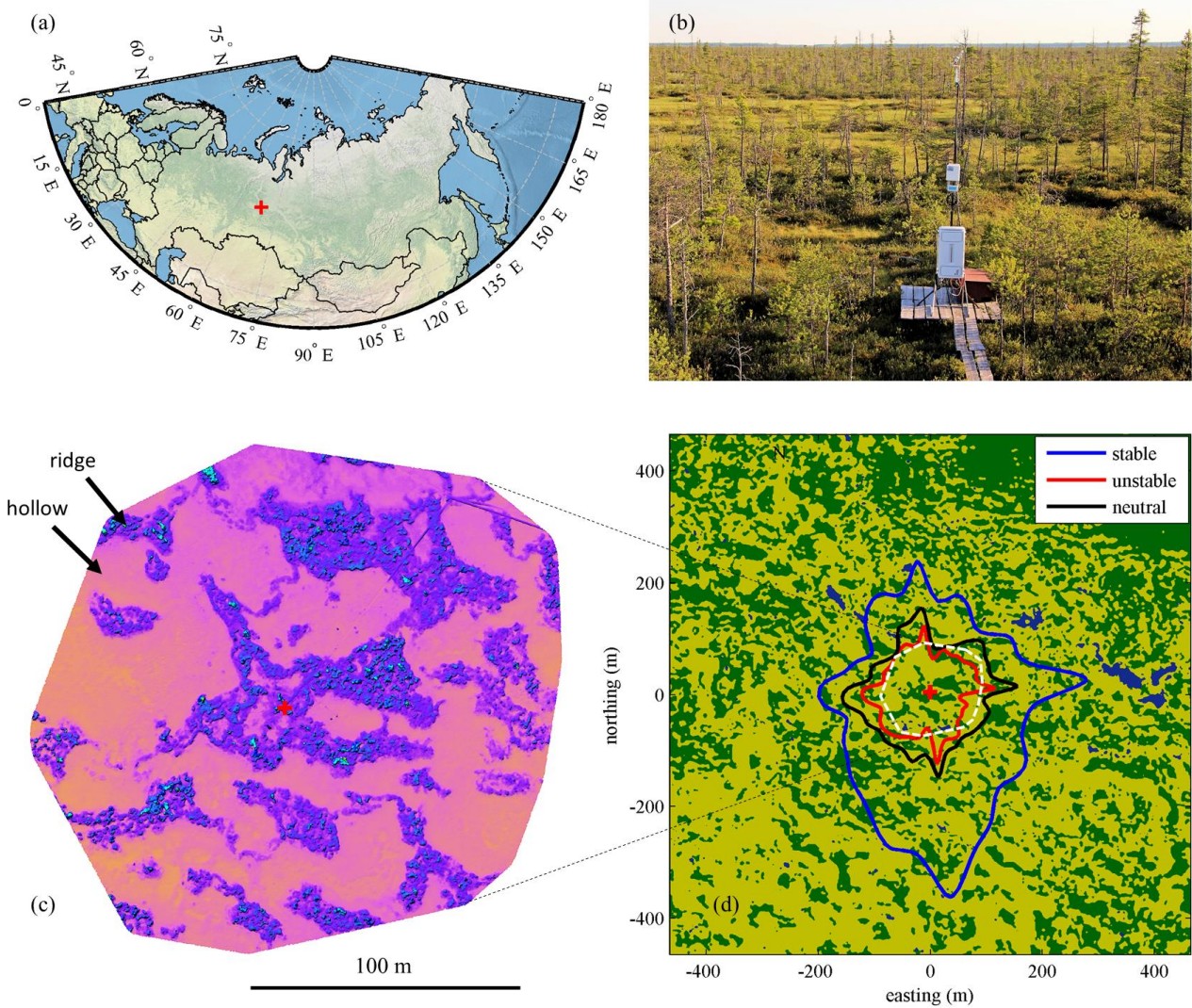

**Figure 1:** (a) map showing the Mukhrino station location, (b) photo of the EC tower facing southwest, (c) digital elevation map based on drone survey, (d) surface type classification map. (d) includes an eddy-covariance footprint overlay, with the isolines giving the 70% cumulative EC source zone in the three stability classes. Color coding in (d): dark green – ridges/hummocks, light green – lawns/hollows, dark blue – ponds. The red cross marks the location of the EC tower.

The Mukhrino Field Station (map, Fig. 1a) is located on the eastern edge of a peatland 10 km × 5 km size. The study site is considered to be representative of raised bogs, a peatland type dominant not only in the west (Masing et al., 2010), but also in the other parts of Siberia (Shulze et al., 2015). The peat layer of up to 5 meters in depth is composed of Sphagnum with minor contributions by other plants. The pH is 3.5-5, electric conductivity – from 0 to 200 $\mu Sm/m^2$ (Sabrekov et al., 2011). The rate of peat accumulation at a nearby wetland site was 0.35 mm $yr^{-1}$, while the average dry bulk density of the peat was 92.7 g $dm^{-3}$ with the average C peat content of 52.7% (Turunen et al., 2001).

Pine bogs and ridge-hollow complexes are dominant within the boundaries of Mukhrino bog (Fig. 1b-d). The tree cover of ridges and pine bogs is represented by stunted *Pinus sylvestris*. The dwarf shrub layer consists of *Ledum palustre*, *Andromeda polifolia*, *Chamaedaphne calcylata*, *Vaccinium vitis-idaea*, *Vaccínium uliginosum*, *Oxycoccus palustris*. Herbs are represented by *Rubus chamaemorus* and a few tiny species of sundews (*Drosera anglica*, *D. intermedia*, *D.*

*rotundifolia). Carex limosa*, *Eriophorum vaginatum*, *Scheuchzeria palustris* are widespread within oligotrophic hollows of ridge-hollow complexes. The moss layer of raised bogs consists of Sphagnum mosses as *S. fuscum*, *S. lindbergii*, *S. balticum*, *S. papillosum*, *S. angustifolium*, *S. magellanicum*, *S. jensenii*, etc. The area fractions of open water, hollows and ridges within a 200 m radius around the flux tower are 1%, 67% and 32%, correspondingly (Fig. 1c-d).

Over the past years, the Mukhrino bog has been in the focus of a large number of studies ranging from surface-atmosphere gas exchange (Glagolev et al., 2011) to geochemistry and physical, chemical and biochemical properties of peat (Stepanova and Pokrovsky, 2011;Szajdak et al., 2016), hydrology (Bleuten and Filippov, 2008), microbiology including mycology (Filippova et al., 2015).

### 2.2 Measurements

Turbulent fluxes of momentum, sensible (H) and latent (LE) heat, and $CO_2$ were measured between 1[st] May and 2[nd] September 2015 with the eddy-covariance (EC) technique. The EC system included a 3-D ultrasonic anemometer (Gill R3, Gill Instruments Limited, Great Britain) providing three wind velocity components and the sonic temperature, and an open-path infrared gas analyzer (LI-7500, LI-COR Biosciences, USA) for the measurement of $CO_2$ and water vapor ($H_2O$). The EC sensors were mounted on a tower at a 4 m height above the peat surface. The horizontal separation between the sonic anemometer and the gas analyzer was 15 cm. The open-path gas analyzer was connected to the analogue input of the sonic anemometer. The data were logged on a mini-computer via serial cable at the sampling frequency of 10 Hz. The eddy-covariance tower coordinates are 60.89133˚ N, 68.67627˚ E.

Auxiliary parameters were measured and recorded by two automatic meteorological stations located within 30 m from the EC tower. The measured parameters include the soil temperature profiles at depths of 2, 5, 10, 20 and 50 cm (thermocouple sensors), net radiation (Kipp&Zonen NRLite radiometer), incoming and reflected photosynthetically active radiation (Li-Cor LI-190SA Quantum Sensor), air temperature and relative humidity (ROTRONIC HygroClip S3) and precipitation (HOBO Data Logging Rain Gauge RG3-M). The soil temperature profiles were installed in ridge and hollow, two replicates in each microsites; the replicates were averaged for further uses. Water table level was also measured in both types of microsites with two Mini-Diver sensors (DI 501).

The station uses an autonomous power supply system consisting of solar panels (4 kW in total) and a wind vane generator (3 kW). A combined charge controller/invertor unit charges the batteries with the total capacity of 800 Ah and supplies up to 3 kW to the field station. In wintertime, a 2.5 kW petrol generator is additionally used.

### 2.3 Flux calculation

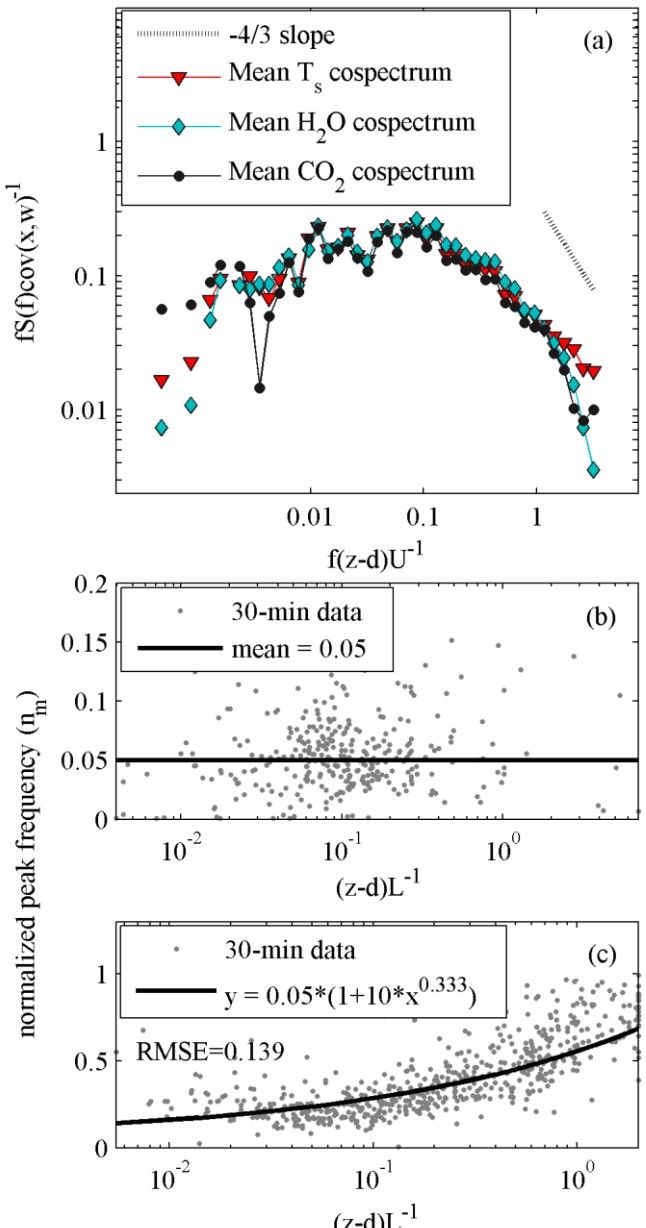

**Figure 2:** (a) Normalized frequency-weighted co-spectra of temperature, carbon dioxide and water vapour as a function of normalized frequency measured on 16 July 2015, 11:00-13:30 (mean value of stability parameter $(z-d)L^{-1}$ = -0.023). The two following subplots show the 30 min values of the normalized peak frequency $n_m$ *versus* the stability parameter $(z-d)/L$ in unstable condition (b) and stable conditions (c).

The post-field processing of EC rawdata was performed with EddyUH software (Mammarella et al., 2016). Fluxes of sensible and latent heat and $CO_2$ were calculated as the 30-min block- averaged covariances between the scalars and the vertical wind velocity:

$$H = \rho_d c_p \overline{w'T_a'} \tag{1}$$

$$LE = \rho_d L_v \frac{M_w}{M_a} \overline{w'\chi_{H_2O}'} \tag{2}$$

$$F_{co_2} = \frac{\rho_d}{M_a} \overline{w' \chi_{CO_2}}'$$

(3)

155

where $\rho_d$ is the dry air density (kg m$^{-3}$), $c_p$ the specific heat capacity of dry air (J kg$^{-1}$ K$^{-1}$), $L_v$ is the latent heat of vaporization for water (J kg$^{-1}$), $T_a$ the air temperature ($K$) and $M_a$ and $M_w$ the molar masses of dry air and water, respectively. The terms $\overline{w'T_a}'$, $\overline{w'\chi_{H_2O}}'$ and $\overline{w'\chi_{CO_2}}'$ are the covariances between $w$ and $T_a$, dry mole fractions of $CO_2$ and $H_2O$, respectively.

160

 Data were de-spiked according to standard methods (Vickers and Mahrt, 1997), thereafter wind velocity components were rotated into a natural coordinate system by performing a two-step rotation to each 30 min interval setting the x axis along the mean wind direction and zeroing the mean vertical wind velocity. The time delay between the vertical wind speed $w$ and the scalar ($CO_2$ or $H_2O$) was derived for each 30 min interval by maximizing the respective cross-correlation function, calculated in a very narrow window (from -0.5 s to 0.5 s). The fluxes were corrected for high and low frequency losses that occur due to the limited frequency response of the EC system and the finite time averaging period used for calculating the fluxes, respectively. Correction was done according to Mammarella et al. (2009) by using experimentally and theoretically determined co-spectral transfer functions at high and low frequency. The estimated low pass filter time constant for $CO_2$ and $H_2O$ was 0.05 s. The effect of this correction is very small, and is mainly caused by the separation between the open-path analyzer and the sonic anemometer. The high frequency attenuation can be clearly seen in the measured co-spectra (Fig. 2a). When performing the spectral correction to the $CO_2$ and $H_2O$ fluxes, the derived transfer functions were used together with the site-specific co-spectral model, which was estimated by a non-linear fit of the measured $\overline{w'T}'$ co-spectrum. The normalized frequency of the co-spectral peak ($n_m$) was also estimated from the co-spectrum for each 30 min record, and the site-specific stability dependence was established (Fig. 2b-c). In unstable conditions (stability parameter z/L<0) $n_m$ has a constant value of 0.05, whereas in stable conditions an increase with atmospheric stability is observed (z/L>0). Before calculating the sensible heat flux, the 30 min sonic temperature covariances are converted to actual air temperature covariances following the approach of van Dijk et al. (2004). LE and $CO_2$ fluxes are corrected for air density fluctuations (Webb et al., 1980). Finally, the Burba correctinon (Method 4 in Burba et al., 2008) was applied to the $CO_2$ and LE fluxes.

180 Ground heat flux (G) was calculated for the ridge and hollow microforms from the peat temperature profile as heat storage change in the top 50 cm of soil following the methodology described in Ochsener et al. (2007) and elsewhere:

$$G = \int_0^{50cm} C_v \frac{\partial T}{\partial t} dz$$

(4)

185

The total volumetric heat capacity, $C_v$, was calculated as a sum of volumetric heat capacities of the solid, water and air constituents, weighted by their volume fractions in the soil matrix. The temperature measurements at 5, 10, 20 and 50 cm depths were used here. Volumetric soil water content profile was modeled as a function of water level (Yurova et al., 2007;Weiss et al., 1998):

190

$$\theta(z) = \phi[1 + (a(-(WT - z)))^b]^{-1+1/b} \tag{5}$$

where $\phi$ is the soil porosity and a, b the empirical parameters. $\phi$ was taken as 95%, corresponding to the average of the representative acrotelm and catotelm values (Granberg et al., 1999). Therefore, the fraction of solid peat particles constituted the remaining 5% of the volume. a and b were adopted from Yurova et al. (2007). The ridge and hollow WT measurements were used to model $\theta$ for two microsite types. Finally, a footprint-representative estimate of G was obtained as an average of the ridge and hollow fluxes weighted by the respective area fractions (68% and 32%).

### 2.4 Flux quality criteria and footprint

In this study, we analyzed the data in the period between 1st of May to 31st of August 2015 (122 days). A long gap in $CO_2$ and $H_2O$ flux data, due to IRGA malfunction, occurred between 25 July and 6 August 2015. Short gaps during night time amount to 73% of the total night time periods, being mainly due to limited power availability, but also low turbulence conditions. Night time was defined as the periods with PAR<10 $\mu$mol m$^{-2}$ s$^{-1}$. Other instrumental problems causing spikes in the measured $CO_2$ and $H_2O$ signals (mainly caused by rain) were eliminated by the despiking method as described in Section 2.3 and by visual inspection of the raw data timeseries. The 30 min time series containing more than 5 spikes were discarded from further analysis, causing a loss of about 4% of $CO_2$ and $H_2O$ data and 13% of sonic anemometer data. Only the 30 min records with friction velocity ($u_*$) larger than 0.1 m s$^{-1}$ and fluxes with stationarity less than 1 (Foken and Wichura, 1996) were used in further analyses. Finally, the overall data coverage for quality-controlled and filtered $CO_2$, sensible and latent heat fluxes during the chosen period was 28%, 33% and 35%, respectively. $CO_2$ nocturnal data of August was affected by spikes and so excluded from analysis.

Flux footprint was estimated using the Kormann and Meixner (2001) model. In the calculations, a value of 0.12 m (an average for May-August calculated from sonic anemometer data assuming a logarithm wind profile in near-neutral stability conditions) for the aerodynamic roughness length was used, whereas wind speed, Obukhov length and standard deviation of lateral wind velocity component were acquired from the EC data. The average source area contributing 70% of the flux ranges from 89 m in unstable conditions up to about 116 m in near-neutral and 202 m in stable conditions (Fig. 1d).

### 2.5 Energy flux gapfilling

In order to avoid systematic bias in calculation of cumulative energy flux values, the energy flux time series were gapfilled. The soil heat flux record was complete, as for its calculation gapfilled peat temperature series were used. The rest of the fluxes were gapfilled individually following Falge et al. (2001). First, the net radiation flux was gapfilled using the mean diurnal variation (MDV) and lookup tables for longer gaps, and with linear interpolation for shorter gaps. Next, LE and H were gapfilled using linear regression against $R_n$. The MDV and linear regression models were calculated in a moving time window 10 days wide. The $R^2$ of the gapfilling models reached 0.68 for H, 0.81 for LE and 0.89 for $R_n$.

### 2.6 Partitioning and gapfilling of Net Ecosystem Exchange

The net ecosystem exchange (NEE) measured by EC was partitioned into ecosystem gross primary production (GPP) and ecosystem respiration ($R_e$), and then gapfilled, following the next steps:

1) A NEE model incorporating a $Q_{10}$-type expression for respiration and a rectangular hyperbolic GPP expression (Eq. 6) was fit to the data at PAR < 300 Wm$^{-2}$.

$$NEE = \underbrace{R_{ref}Q_{10}^{\left(\frac{T_0-T_{ref}}{10}\right)}}_{a)\ R_e} - \underbrace{\frac{P_{max}PAR}{k+PAR}}_{b)\ GPP} \tag{6}$$

where $T_0$ is the area-weighted average temperature of hollows and hummocks at a 5 cm depth (˚C), $T_{ref}$ the reference
temperature of 12˚C, $R_{ref}$ the reference respiration ($\mu mol(CO_2)$ m$^{-2}$ s$^{-1}$), and $Q_{10}$ the temperature sensitivity, PAR the photosynthetically active radiation ($\mu mol(CO_2)$ m$^{-2}$ s$^{-1}$), $P_{max}$ the maximum photosynthesis ($\mu mol(CO_2)$ m$^{-2}$ s$^{-1}$), k the value of PAR at $1/2P_{max}$ ($\mu mol(CO_2)$ m$^{-2}$ s$^{-1}$). All the four parameters ($Q_{10}$, $R_{ref}$, $P_{max}$ and k) were evaluated by fitting at this step. The resulting $Q_{10}$=1.99 (95% CI [1.42; 2.57]) was then fixed for the whole study period.

2) The respiration module (a) of Eq. (6) was fit to the nighttime data in a 30-day-wide moving time window, with $R_{ref}$
being evaluated at each step. The window was shifted by 1 day steps.

3) In the same time window, GPP was calculated as the residual of measured NEE and respiration model, after which the GPP module (b) of Eq. (6) wat fit to produce the values of $P_{max}$ and k.

4) The daily "window" values of $R_{ref,}$ $P_{max}$ and k were then smoothed out with the spline interpolation procedure and rescaled down to the 30-minute resolution of the original data. Fig. 3 shows the resulting parameter time series.

5) Finally, the $R_e$ and GPP models were calculated at a 30-min resolution and used to fill the gaps in the measured NEE.

## 3. Results and discussion

### 3.1 Environmental conditions

Weather in Mukhrino during the summer season of 2015 was unusual for the regional climate. The spring was early and
warm: the average air temperatures in May and June were 4.1 °C, or 3.4 °C higher than the long-term average (Fig. 3a). It caused an unusually early and rapid snowmelt in April and the beginning of May; the last patches of snow melted by 3$^{rd}$ May, while three temperature profiles out of four indicated freezing until 3$^{rd}$ May at -5 cm and until 6$^{th}$ May at -20 cm depths. In contrast to the climatic average, June was the warmest month with an average air temperature of 18˚C and a maximum value of 32˚C. However, the rest of summer was greatly affected by the cool fronts that brought
precipitation and cloudiness. For this reason, the average July and August values sunk below the average by 2.7 and 1.7°C, respectively. Maximum soil temperature at a 20 cm depth (19 °C) was observed in the last decade of June, while soil temperature at 50 cm had two maxima of 16°C in beginning of July and in first decade of the August (Fig. 3a). Photosynthetically active radiation was at its maximum in May-June, slightly decreasing in July and August (Fig. 3c). The maximum midday value of 1463 $\mu mol$ m$^{-2}$ s$^{-1}$ was registered in the middle of June.

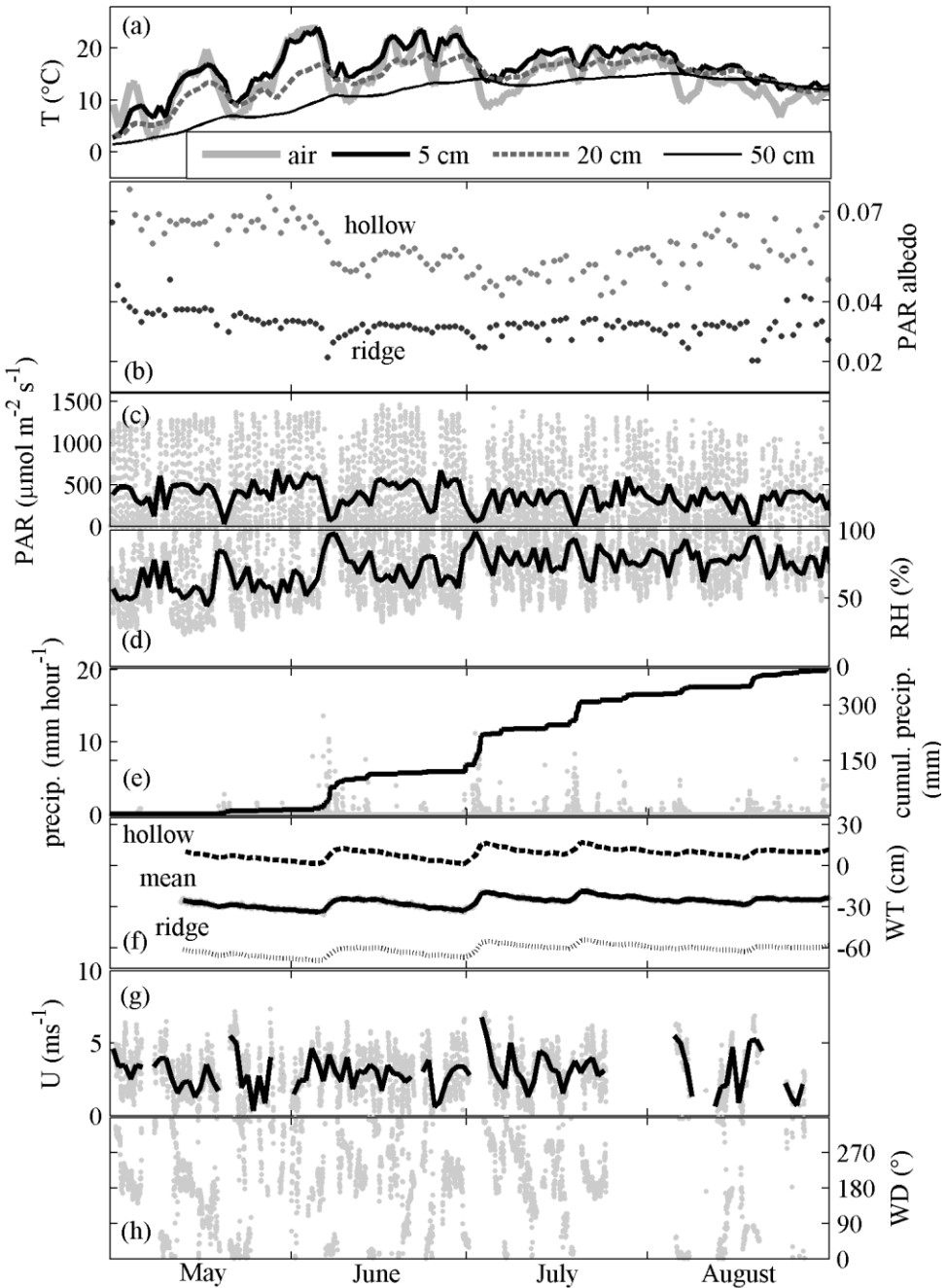


**Figure 3:** Time series of the environmental variables: (a) air and peat temperatures, (b) PAR albedo, (c) PAR, (d) relative humidity, (e) precipitation, (f) water table depth, (g) wind speed and (h) wind direction. The grey dots are 30 min measurements, while the lines represent daily averages, except in (b) where the dots are midday (10:00-16:00) medians of PAR albedo and in (e) where a bold line shows the cumulative precipitation. In (a), the presented peat

temperatures are area-weighted averages of 2 hollow and 2 ridge measurement locations.

Precipitation considerably differed from the 81-year reference period (not shown). It was 2.7 times higher in June-July because of three heavy rainfall periods (4-9 June, 2-5 July, 18-20 July 2015). The total cumulative precipitation of the study period (May-August) was 405 mm, or 45% higher compared to the reference period, and 325 mm in May-July

(Fig. 3e). The frequent precipitation helped to sustain high relative humidity (d). Accordingly, the water table depth (WTD) changes followed the intensity and frequency of precipitation, decreasing slowly during dry periods, and rapidly

increasing in heavy rain (Fig. 3f). The snow resided on the ground until 3$^{rd}$ May and after 30$^{th}$ September. In fact, the end of snowmelt can be seen as a steep reduction in PAR albedo in the beginning of May (Fig. 4b; the hollow albedo starts at 0.12 in early May, but the axis is limited at 0.08 for clarity). Otherwise, PAR albedo follows the typical trends, being higher in the hollows than in the ridges and showing downward peaks during the precipitation events.

The prevailing wind direction was from the South/South-West (150-260˚, 45% of cases), in which the proportions of open water, hollows and ridges within the 200 m radius are 1%, 67% and 32%, correspondingly. Similar proportions hold for the entire area within a 200 m radius around the mast.

**3.2 Surface energy exchange**

Time series of the surface energy fluxes and their monthly diurnal courses are shown in Figs. 4 and 5, respectively. The midday net radiation ($R_n$) averaged 397 W m$^{-2}$ and 364 W m$^{-2}$ in May and June, respectively, reflecting the large amounts of incoming solar radiation (Fig. 3c). However, the high post-snowmelt water levels and undeveloped vascular vegetation in May resulted in low albedo, somewhat lowering $R_n$. On the other hand, the midday values in the second part of the summer are clearly lower, being 275 W m$^{-2}$ and 211 W m$^{-2}$ in July and August, respectively. In fact, as later in the summer the developed ground vegetation attains a higher reflectivity, this increases the surface albedo and decreases $R_n$. In addition, the frequent overcast conditions (16 days in July and 17 in August) further reduced incoming solar radiation in late summer (Table 1). The soil gained heat throughout most of the studied period, but the average flux is very small, ranging from 8 W m$^{-2}$ in May to 1 W m$^{-2}$ in July. The ground starts to cools down in August with G equaling -4 W m$^{-2}$. Most of the available energy is released as latent heat flux, whose monthly average values are between 61 and 96 W m$^{-2}$, while sensible heat fluxes are more than three times lower ranging from 30 W m$^{-2}$ in July to 13 W m$^{-2}$ in August (Table 1).

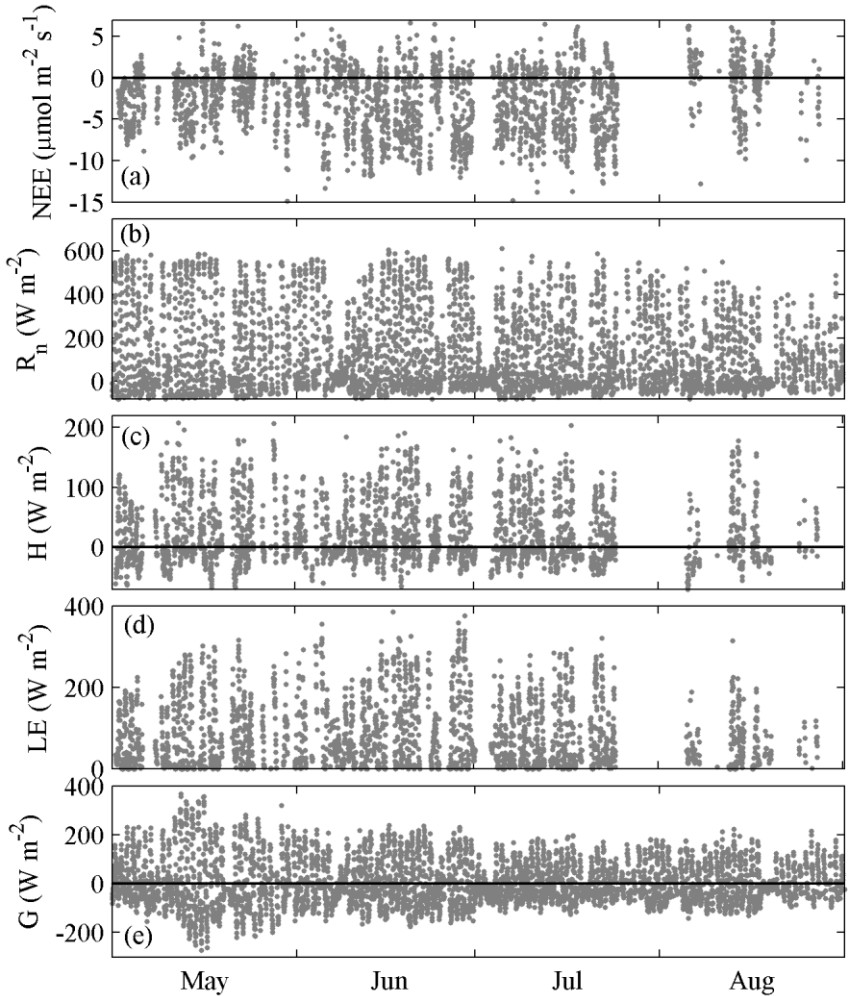

**Figure 4:** Time series of the 30-minute average surface fluxes measured with the eddy-covariance system: (a) net ecosystem exchange, (b) net radiation, (c) sensible heat, (d) latent heat, and (e) ground heat flux.

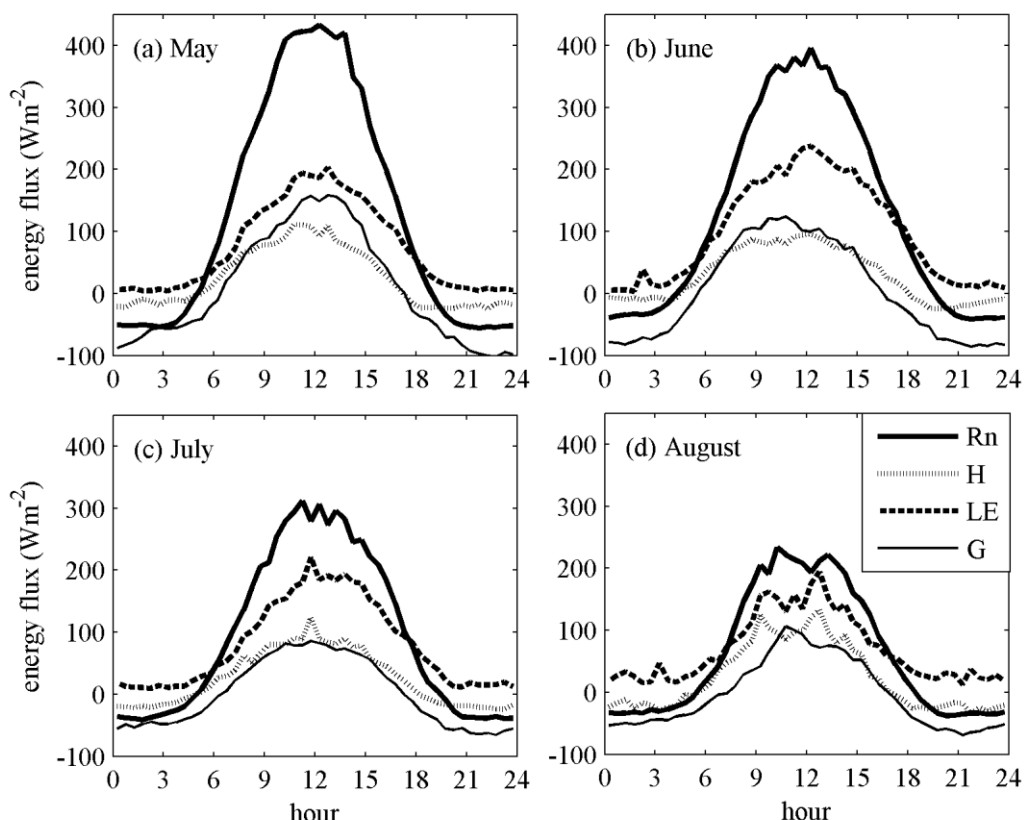

**Figure 5:** Monthly average diurnal courses of the energy balance components. The time shown in the x-axis is local winter time (UTC+5).

There is a marked seasonal change in LE, which starts to increase rapidly in May due to high $R_n$ values, reaching the daily mean peak in July (239 W m$^{-2}$), and then decreases in July reaching the minimum value at noon of about 140 W m$^{-2}$ in August (Fig. 5 and Table 1).

**Table 1.** Monthly averages of air temperature ($T_a$), soil temperature at 5 cm depth ($T_p$), photosynthetic active radiation (PAR), cumulative precipitation (mm), net radiation ($R_n$), ground heat flux (G), sensible heat flux (H), latent heat flux (LE), Bowen ratio (ß), energy balance residual (Res = $R_n$-H-LE-G), energy balance closure (EBC = (H+LE+G)/($R_n$)), and gapfilled NEE.

| | $T_a$ [°C] | $T_p$ [°C] | PAR [Wm$^{-2}$] | P [mm] | WTD [cm] | $R_n$ [Wm$^{-2}$] | G [Wm$^{-2}$] | H [Wm$^{-2}$] | LE [Wm$^{-2}$] | ß [-] | Res [Wm$^{-2}$] | EBC [-] | NEE$_{gapf}$ [gC m$^{-2}$] |
|---|---|---|---|---|---|---|---|---|---|---|---|---|---|
| May | 11.1 | 11.8 | 405 | 12 | -17 | 126 | 8 | 23 | 73 | 0.32 | 22 | 0.87 | -35 |
| June | 17.9 | 19.6 | 411 | 121 | -16 | 133 | 3 | 30 | 96 | 0.27 | 5 | 1.04 | -72 |
| July | 15.2 | 18.0 | 318 | 191 | -11 | 88 | 1 | 15 | 69 | 0.27 | 3 | 1.01 | -79 |
| August | 12.6 | 15.2 | 295 | 80 | -13 | 62 | -4 | 13 | 61 | 0.26 | -8 | 1.27 | -16 |
| May-August | 14.3 | 16.1 | 359 | 405 | -14 | 102 | 2 | 20 | 74 | 0.28 | 6 | 0.99 | -202 |

Although a seasonal change of H is also observed, it is characterized by a smaller amplitude. Monthly mean values of Bowen ratio (ß) are rather low (around 0.3), showing no significant seasonal variation (Table 1). However, the sequence

of rainy and dry periods caused short-term variations of ß between 0.15 and 0.6 on a timescale of about 2 weeks. Turbulent heat fluxes (H and LE) show a diurnal variation typical of land ecosystems, being in phase with $R_n$ (Fig. 5). The dominance of LE (with respect to H) for northern wetlands has been already reported. Using a flux tower in a Western Siberia bog site located close to Plotnikovo (56˚51' N, 82˚50' E), Shimoyama et al. (2003) show values of ß ranging from 0.57 in the early growing season to 0.78 in the peak growing season. Similar values were found by Aurela et al. (2015) in a Finnish wetland (ß = 0.78, Lompolojänkkä) and in the wetland site Degerö in Sweden (ß = 0.83, Peichl et al. (2013)). However, lower values of ß, more similar to the one in our study, were observed in other northern wetlands (Runkle et al., 2014;Wu et al., 2010;Eugster et al., 2000). Most probably, the difference can be explained by difference in water table depths. One has to account also for the unusually wet conditions in 2015, which must have enhanced LE to a certain extent. High water availability supported high LE and G in May and June. The subsequent reduction in G could have been partly related to higher ground shading by aboveground biomass.

The energy balance closure (EBC) for the whole period is around 1.07 (Fig. 6) when adopted as a slope of LE+H+G vs. $R_n$, or 0.99 when calculated as a ratio of the corresponding cumulative fluxes. The slight excess on the side of H+LE+G might be the product of uncertainty in the modeled peat water content (Eq. 5), which may in turn have affected G. Monthly estimates of the EBC and the residual (Res) are reported in Table 1. The EBC (ratio method) ranges from 0.87 in May, when the difference between available energy and the turbulent fluxes is 22 W m$^{-2}$, to 1.27 in August, when Res is -8 W m$^{-2}$. The summer months (June and July) show the best values of EBC and Res. The observed values are in line with those from other wetland studies (Kurbatova et al., 2002;Peichl et al., 2013;Runkle et al., 2014). In their FLUXNET site-based energy balance closure study, Stoy et al. (2013) reported an average value of 0.76 for the wetland site category, highlighting the relevance of including the heat storage term in sites with high water table depth. Shimoyama et al. (2003) obtained a better EBC value (0.9 vs 0.82 in July), estimating the soil heat flux in the bog from an area-averaged value of soil thermal parameter (instead of using a point value), to some extent accounting for the surface heterogeneity and the presence of microtopography. Our use of area-weighted  average temperature profiles and individual water level measurements in the ridge and hollow microsites has resulted in a similarly high EBC, implying that the main component of spatial heterogeneity must have been captured.

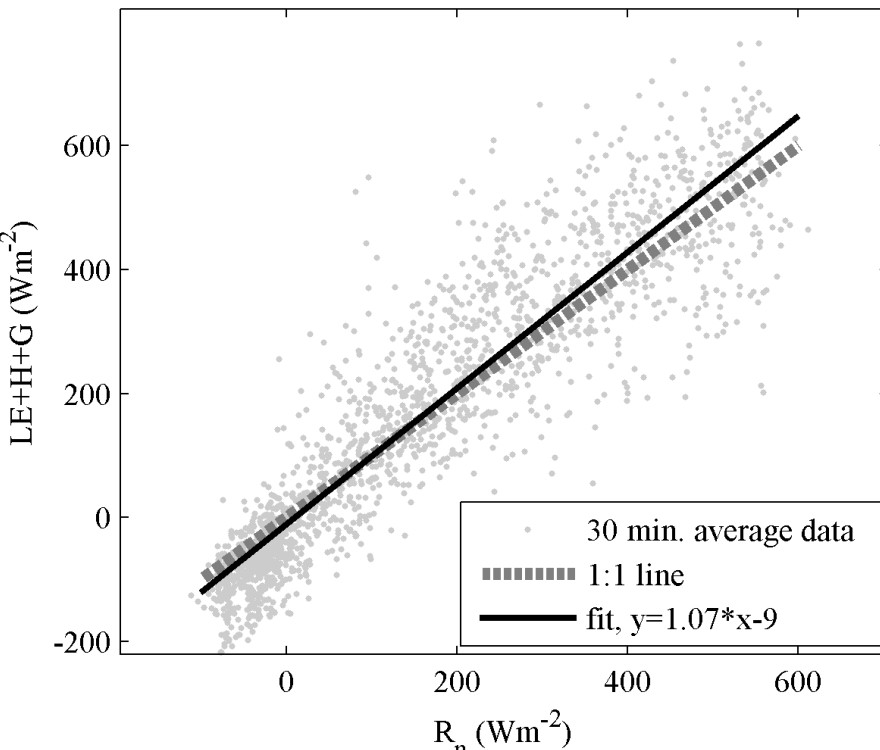

**Figure 6:** Energy balance plot. The sum of latent and sensible heat fluxes and soil heat flux is plotted against net radiation flux. A linear function is fit to the data and shown together with a 1:1 line. The mean ratio of LE+H+G to $R_n$ is 0.99.

340

### 3.3 Carbon exchange

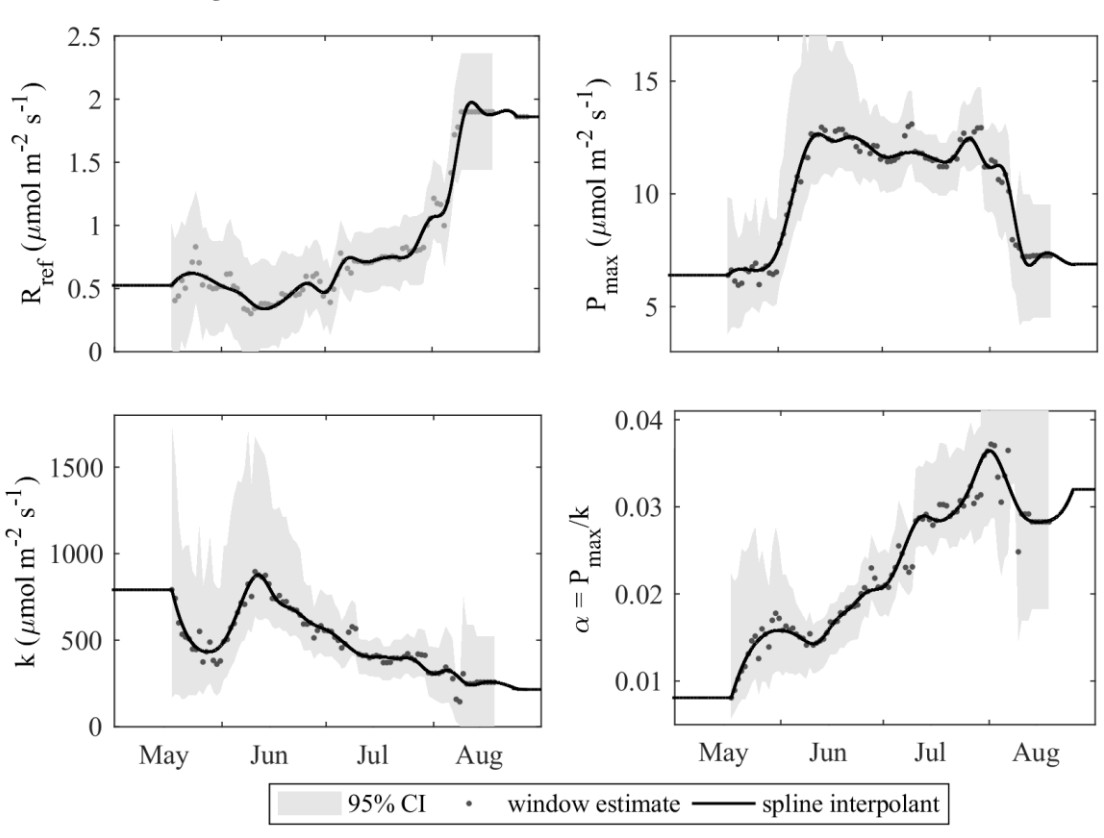

**Figure 7**: time series of the $CO_2$ flux model parameters. The dots are the daily values evaluated in a moving time window, and the solid line is a spline-interpolant, while the shaded area shows the 95% confidence interval, which is calculated at each time window step. The interpolated parameter lines stretch to the beginning of May and end of August, showing the constant values used on the edges of the study period.

The time series of $P_{max}$, k, quantum yield $\alpha$ and $R_{ref}$ presented in Fig. 7 reveal notable trends. $R_{ref}$ drops between May and early June, possibly related to rainy weather spells, but then increases rapidly by August, probably in response to the increasing soil temperatures, availability of substrate and plant productivity. $P_{max}$ has a broad peak in June-July, which is probably governed by the vascular leaf area index, with a possible contribution of the acclimation, as the air temperatures were closer to optimal in that period (see Fig. 10 below). k had a maximum in June, with a later gentle reduction towards August. This evidence of higher photosynthetic activity at low light in late summer may again point at acclimation. In turn, the $P_{max}$ and k evolutions result in a May-August upward trend in $\alpha$. However, one could speculate that this behavior is due to the seasonality of plant functional group activity, to an extent. A Finnish boreal fen study of Korrensalo et al. (2017) found a wide seasonal variation in the contributions of moss and vascular species to the ecosystem-scale photosynthesis. The moss photosynthesis ($gC$ $m^{-2}$ $d^{-1}$) declined steadily throughout summer, while various vascular species were most active in June, July or August; it was also common for k of many species to have a peak in June and/or become reduced throughout the growing season, in line with the findings of the current study.

The time-series of gapfilled NEE is shown in Fig. 8a. The largest flux amplitudes are found during June and July when half-hour values range between about -10 $\mu mol$ $m^{-2}$ $s^{-1}$ and 5 $\mu mol$ $m^{-2}$ $s^{-1}$. In May, the measured NEE has a narrower amplitude because of lower temperature and PAR values. Monthly differences in the NEE amplitude can be clearly seen in Fig. 9, where the mean diurnal course is plotted for each month. The modelled NEE (Eq. 6) closely follows the measured NEE. As expected, the largest net carbon uptake was observed in June and July (-72 and -79 $gC$ $m^{-2}$, respectively), while in May it amounted to -35 $gC$ $m^{-2}$. Unfortunately, poor data coverage was achieved in August, making the corresponding monthly cumulative value somewhat more uncertain, although it seems to have been much lower (-16 $gC$ $m^{-2}$). Overall, the bog site acted as a net $CO_2$ sink in the analyzed period. The cumulative gap-filled NEE for of May-August was -202 $gC$ $m^{-2}$, which decomposes into 157 $gC$ $m^{-2}$ of $R_e$ and -364 $gC$ $m^{-2}$ of GPP. For the period of May-July with the best data coverage, the corresponding values were -186, 91 and -277 $gC$ $m^{-2}$. The Mukhrino May-August GPP falls between 224-243 $gC$ $m^{-2}$ observed over the same period by two Canadian bogs and 466-539 $gC$ $m^{-2}$ in Mer Bleue (Humphreys et al. (2014)). However, in terms of net summer uptake, the Mukhrino was among the highest estimates. For example, Friborg et al. (2003) reported an average July $CO_2$ uptake of -7545 $mg$ $m^{-2}$ $d^{-1}$ (which corresponds to a cumulative sum of -64 $g$ $C$ $m^{-2}$) measured with EC at the Bakchar bog close to Plotnikovo village in West Siberia. This value is very close to the Mukhrino NEE sum of June (Table 1). Lower $CO_2$ uptake is reported for Zotino bog, where growing season (May-October) cumulative NEEs range between -43 and -60 $g$ $C$ $m^{-2}$ in different years, with maximum daily mean NEE of about -2 $\mu mol$ $m^{-2}$ $s^{-1}$ measured in July 2000 (Arneth et al., 2002). Lower NEEs in the range of -50 to -90 $gC$ $m^{-2}$ $year^{-1}$ were shown for several peatlands of Northern European Russia and Siberia (Dolman et al., 2012). Daily net carbon uptakes ranging between -1 and -2.8 $g$ $C$ $m^{-2}$ $d^{-1}$ were measured during summer in other northern peatlands in Canada (Humphreys et al., 2006); compare with the June-August average of -1.8 $g$ $C$ $m^{-2}$ $d^{-1}$ in Mukhrino.

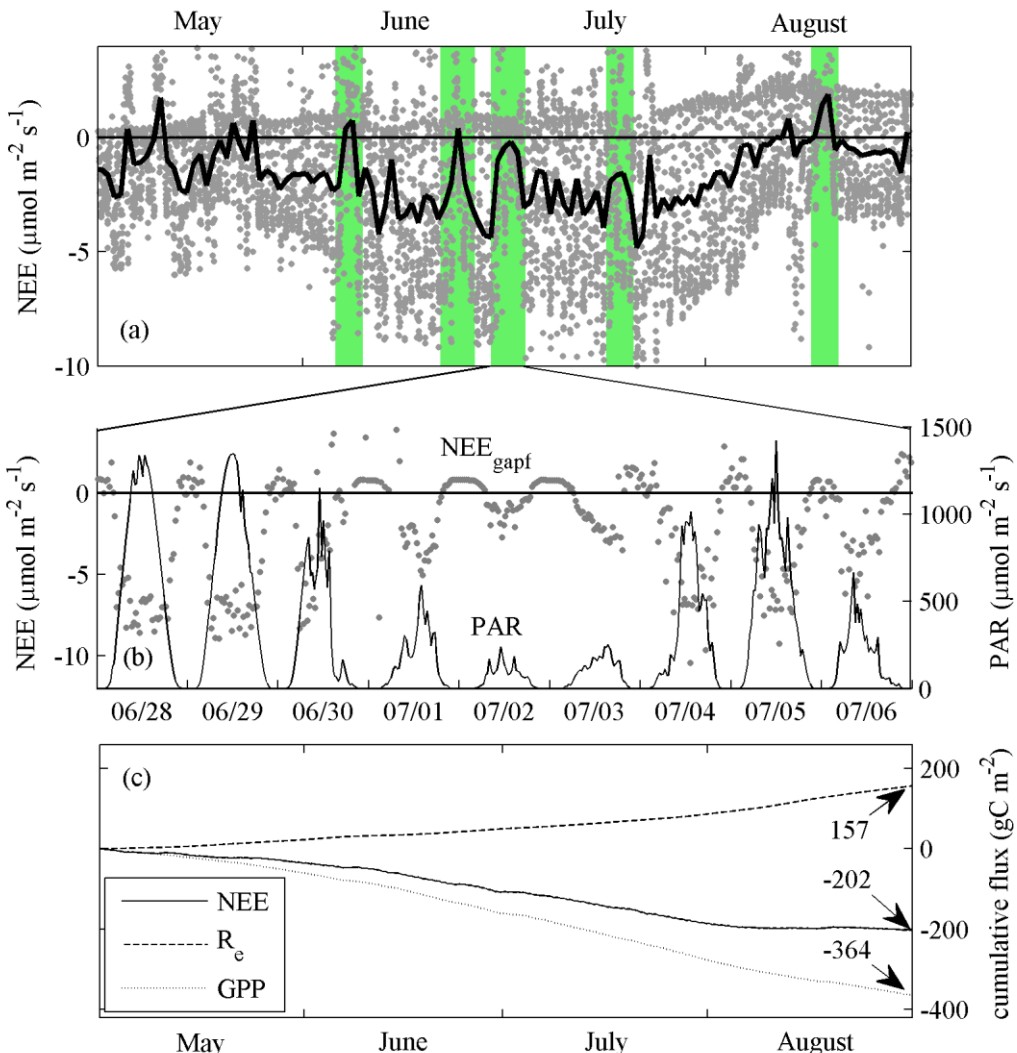

**Figure 8:** (a) Seasonal variation of NEE measured with the eddy-covariance system; the grey dots correspond to 30 min averages and the black line to the daily averages. The major front passage events are marked with green. (b) Measured NEE and PAR during the third frontal event. (c) Cumulative gapfilled NEE, $R_e$ and GPP.

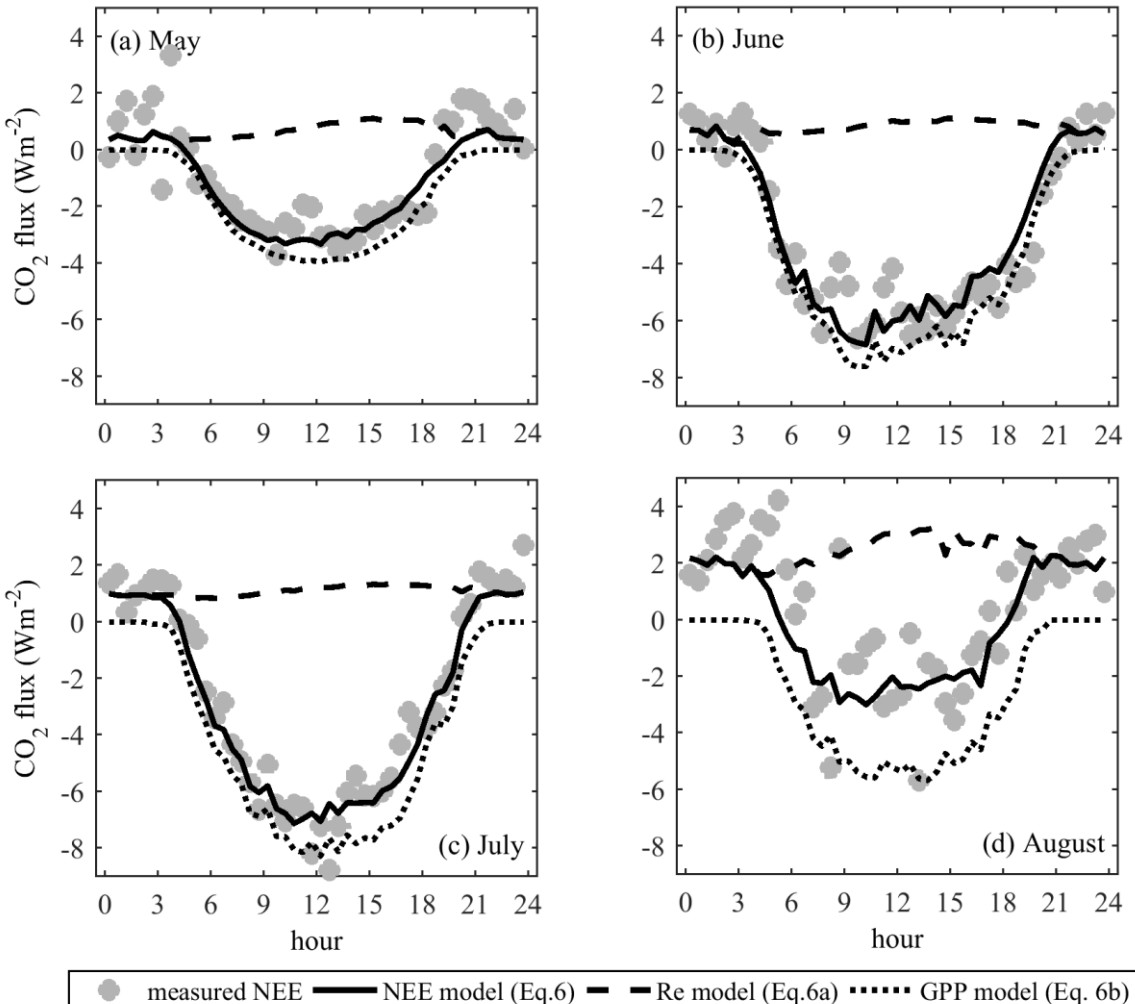

**Figure 9:** Mean diurnal course of NEE and its components for individual months during the study period. The time in the x-axis corresponds to local winter time (UTC+5).

### 3.4 The effect of weather conditions

390    The year 2015 was characterized by diverse weather, starting with an early and warm spring and continuing rainy and overcast. Carbon uptake became significantly limited during the passage of five cold fronts that occurred in June-August (see Fig. 8), four of which were associated with ample precipitation. During those short periods, uptake plunged to very low values, with the ecosystem even becoming $CO_2$-neutral or a small source during some periods. A closer look at one such period is provided by Fig. 8b. Excluding the rain-free front in late June, each event brought about 100

395    mm of precipitation, causing WT raises of 8-11 cm (Fig. 3 d,e). However, no dependency of surface exchange on WT was found. The regular and ample precipitation helped sustain water level at a nearly constant level, which was about -60 cm in ridges, while many hollows stayed inundated (Fig. 3f). In a landscape dominated by ridges (in terms of green biomass), drawdown in WT leads to its decoupling from the hydrological state of surface peat (Price et al., 2003), and, therefore, all vegetation functions including photosynthesis. Sustaining high water level must prevent water stress in the

400    hummock vegetation, which constitutes a significant fraction of green biomass in Mukhrino. In such exceptionally wet conditions as in 2015, the top peat must have stayed moisturized most of the time, meaning that water availability was not the limiting factor for $CO_2$ uptake. At the same time, the hollows stayed largely inundated and as such probably made a smaller contribution to photosynthesis than hummocks.

The overcast conditions during front passage also resulted in temperature drops by up to 13˚C. This obviously limited respiration, but also restricted photosynthesis as the optimum growth temperature seems to be close to 30 ˚C (Fig. 10b). High relative humidity, and thus lower VPD, must have partly compensated for the lower $CO_2$ uptake during the fronts by promoting higher stomatal conductance ($g_s$). In terms of the parameters $g_1$ and $m$ (Fig. 10a), the response of $g_s$ to VPD was similar to that in southern Swedish bog Fäjemyr, northern Swedish fen Degerö and southern Finnish fen Siikaneva (Alekseychik et al., unpublished data; Peichl et al., 2013). Mean bulk surface resistance ($r_s$, the reciprocal of conductance) of 74 s m$^{-1}$ is somewhat lower than approximately 85 s m$^{-1}$ reported for the "wetlands" ecosystem class by (Kasurinen et al., 2014). As mentioned above, the Bowen ratio was stable throughout the summer (~0.3), but the weekly mean values varied between 0.15 and 0.6 in close correlation with the precipitation and cloudiness pattern.

Nevertheless, the favorable conditions of May and early June allowed for apparent rapid accumulation of green biomass. Also, between the cold fronts, air temperatures did occasionally exceed 25 ˚C periodically, promoting photosynthesis. This behavior reflects the temperature control on GPP that is common for the whole Boreal region (Reichstein et al., 2007). As a result, the early spring and sustained wetness for the rest of the year seem to have outweighed the GPP restriction during the front passages, and eventually led to an unusually high cumulative $CO_2$ uptake.

No flux variation with wind direction was found, in consistency with the similarity of ridge-hollows fractions in different wind direction sectors (Fig. 1 c,d).

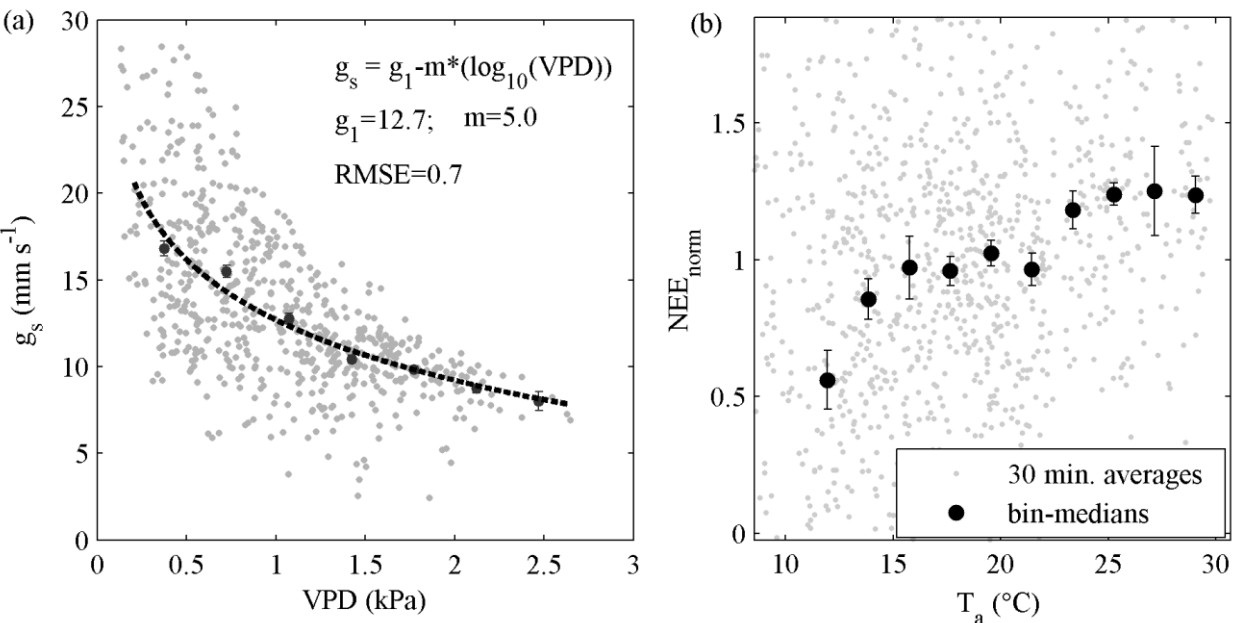

**Figure 10:** (a) Surface conductance *versus* vapor pressure deficit demonstrating a well-known relationship. The dashed line is a fit (the function is specified in the insert). (b) Normalized NEE (NEE$_{norm}$ =NEE/NEE$_{mod}$) *versus* air temperature for June-July 2015; NEE$_{norm}$ saturates approaching 25 ˚C.

## 4. Conclusions

This study provides the results of direct and continuous measurements of surface energy balance components and $CO_2$ flux at the Mukhrino bog site in West Siberian middle taiga. The turbulent fluxes measured by the EC technique over May-August 2015 form a pioneering dataset of its kind for the region.

The observed magnitudes and diurnal course of sensible and latent heat fluxes were generally in agreement with previous bog studies. The latent heat flux was about three times larger than the sensible heat, and the monthly mean Bowen ratio did not show any significant seasonal variation. However, short-term variations related to heavy rainfall events were observed. In terms of monthly averages, May and June were characterized with the highest available energy.

Carbon dioxide exchange was typical of a raised bog, with net $CO_2$ sink being rather high (202 gC m$^{-2}$ for May-August) but within the range of previous observations (IPCC, 2014, 2013). Remarkably wet weather of 2015 ensured high moisture availability and thus promoted high photosynthesis during the sunny periods. However, the rainy and cool conditions during the passage of several fronts limited photosynthesis so that the ecosystem temporarily turned into net $CO_2$ source. The peak in carbon uptake lagged the maximum available energy by one month, falling on June-July, probably being modulated by the course of vascular plant leaf area development.

The sharp seasonality of the photosynthesis and respiration model parameters pointed at an ensemble of effect, including the variability in green biomass, relative importance of plant functional groups, and acclimation. Complementary chamber and plant-scale studies will help disentangle those effects.

The Mukhrino station was established for the purpose of long term monitoring of ecosystem functioning and greenhouse gas exchange and continued its operation in 2016. Obtaining a measurement record over several years with varying weather would be instrumental for determining the typical budgets of the ecosystem, unaffected by untypical weather, which was the case in 2015.

**Author contributions**

P. Alekseychik analysed the data, produced the figures and contributed to the text. I. Mammarella wrote a major part of the text. D. Karpov and S. Dengel set up the eddy-covariance measurements, provided technical support and contributed to the text. I. Terentieva, A. Sabrekov and M. Glagolev participated in the writing process. E. Lapshina supervised the Mukhrino field station.

**Acknowledgements**

The study was supported by INTERACT project GHG-FLUX, EU-project GHG-LAKE (project no. 612642), Nordic Centre of Excellence DEFROST, National Centre of Excellence (272041), ICOS-FINLAND (281255) and CarLAC (281196), funded by the Academy of Finland, and the Russian Foundation for Basic Research grant №15-05-07622. We gratefully acknowledge Dr. Pasi Kolari for the valuable advice he provided. The help in the field and provision of data by Yaroslav Solomin and Prof. Martin Heimann is gratefully acknowledged. Nina Filippova compiled the meteorological data. The codes and data used in this study are available upon request.

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
