# Peer review of "Net ecosystem exchange and energy fluxes measured with eddy covariance technique in a West Siberian bog"

_Atmospheric Chemistry and Physics, 2017_

## Referee Comment (RC1) · C. Wille (Referee) · 13 Mar 2017

General Comments

The manuscript presents energy and CO2 flux data from the West Siberian Taiga. This is valuable data, as the West Siberian Lowland is a vast understudied region. The presented 4-month data set is the first data of what is to become a permanent flux measurement site. Thus is can provide a base line for comparison with other sites and with data that will be collected at the same site in the years ahead.

Generally, the style of the manuscript and the presentation of data is adequate. However, the data analyses lag behind the state of the art and the discussion of the results is often weak. Extensive revisions are necessary before publication of the manuscript.

[Figure]

Main Critique

(1) No information on the gap-filling of energy fluxes is given. Was gap-filling not performed for H and LE? Monthly means of these values could be seriously biased if they are calculated based on non-gap-filled time series. Gap-filling should be performed in order to derive sound estimates of mean or cumulative fluxes, and the methods used should be clearly presented in the methods section.

(2) How have the authors addressed the heterogeneity of soil and hydrological properties and hence ground heat flux (G)? Was soil temperature measured and G calculated for only one microform, hummock or hollow? As G could be expected to vary strongly between hummocks and hollows, a weighted average (based on surface area fractions) of G calculated for both microforms should be used. If G is available for only one microform, an estimate of the error induced by this approach should be added (which could also serve as a justification for this approach).

(3) In my opinion, an instationarity test (e.g. Foken and Wichura, 1996) is start of the art and should be applied.

(4) No information on the seasonal vegetation development is given in the manuscript. Even if assessments/measurements of GAI or LAI are not available, a general description of the vegetation development is indispensable in order to put the observed flux data in context to the annual cycle of fluxes and drivers. Additionally, as the measurements started directly after the end of snow melt, information on the snow/soil conditions immediately before the beginning of the measurement period should be added, if possible (snow height, snow water equivalent, beginning of snow melt, depth of frozen peat layer, beginning/end of peat thaw). This could be very helpful to understand the temporal development of fluxes at the beginning of the growing season.

(5) The partitioning of measured net ecosystem exchange (NEE), particularly the modelling of ecosystem respiration (Re) appears to be not sound. In Detail: (a) Why are there significant negative fluxes in the Re vs. peat temperature (Tp) plot (Fig. 3a)?

[Figure]

After careful QA/QC I would ideally expect to see only few and small negative night time $CO_2$ fluxes (predominantly at lower temperatures). Maybe, the application of an instationarity test could help removing these conspicuous data points? (b) The fit of eq. 6 to the Re vs. Tp data set (Fig. 3a) seems to have a low $R^2$. I'd like to see the $R^2$ and p values for this fit. (d) Generally, combining data from the period May-August in one fit of Re vs. Tp is likely to confound the seasonal development of Rref with its temperature dependence. This is reflected in the large temperature sensitivity (Q10 value) obtained by the fit. Fitting Re vs. Tp in a moving window of length 10...30 days would be more appropriate. If this would lead to unrealistic variations of the reference respiration (Rref) and Q10, the authors could constrain Q10 to a value around 1.5 (cf. Mahecha et al., 2010). This way, at least the variation of Rref could be assessed, which could give valuable insights into the seasonal vegetation development.

Specific Comments

Line 123: On which micro-form was soil temperature measured (hummock/hollow)? See (2) in Section "Main Critique" above.

Line 169: For which micro-form was ground heat flux calculated (hummock/hollow)? See (2) in Section "Main Critique" above.

Lines 193-194: Why was only $CO_2$ night time data of August excluded from analysis? It is hard to imagine that only $CO_2$ fluxes are compromised by technical problems of the gas analyzer but not LE fluxes.

Lines 205-206: Why was night-time defined as periods with a solar elevation angle below $5°$ and not by a PAR threshold (e.g. PAR < 20 $\mu$mol/m2/s)? Using a local PAR threshold may allow additional data points to be included into the night time data set (e.g. during cloudy conditions), which could improve the data coverage and hence the modelling of Re.

Line 206: Have you tried to use peat temperature from other depths or air temperature for the modelling of Re? Information on the performance of the model with other temperatures could give valuable insights into the source of respired CO2.

Lines 219-221: In which time steps was the 30-day window moved? Please add this information. Further, the time series of the fit parameters Pmax and k (or the often used alpha = Pmax/k) should be presented. This could deliver valuable information on the seasonal development of the vegetation and could be compared to other studies.

Lines 234-236: Soil temperature at depths 20 cm and 50 cm is discussed here, but this data is neither displayed in Fig. 4a nor used in the analyses. I suggest to either add this data in an additional subplot of Fig. 4 or concentrate in the text on the data already displayed in Fig. 4, i.e. Tp at 5 cm depth.

Lines 258-259: The statement "...later on during the summer the water level decreases..." contradicts what is shown in Fig. 4e, and is stated in lines 344-345, "The regular and ample precipitation helped sustain water level at a nearly constant level...". Hence, the authors' assumption that albedo is reduced due to drying of the vegetation is ill-conceived. Still, it could be checked by simply calculating an albedo from incoming and reflected PAR.

Line 302: The spatial heterogeneity does not seem to serve as a good explanation for the low value of the energy balance closure in May, as the surface heterogeneity does not change during the course of the measurement period. Or does it change? How?

Line 310 and Fig. 8a caption: The data displayed is surely modelled NEE and not measured NEE?

Lines 311-312: Could the lower amplitude of NEE in May also be due to a not fully developed foliage of the vegetation? Snow melt had only ended a few days before and below-zero temperatures still seem to occur during May. Time series of the parameters Pmax, k, and Rref could help to explain the variations in observed NEE.

Lines 213-314: I see a systematic difference of measured and modelled NEE in Fig.

9. In the afternoon hours of July, measured NEE uptake is smaller than modelled NEE uptake. Hence, either Re is underestimated or GPP is overestimated. What could be the reason for this? Furthermore, why is August night time CO2 flux data displayed if it should have been excluded from analysis due to technical problems (line 194)? In fact, this data does not look completely unrealistic to me. Gažovič et al. (2013) has observed the highest Re during August, while GPP peaked in July. The discrepancies between modelled and measured fluxes could be caused by the fact that Re is poorly modelled by the approach chosen by the authors.

Lines 352-353 and Fig. 10: Combining all data from the period May-August potentially confounds the seasonal development of Pmax and k, and hence GPPmod, with a possible short term variation of these parameters due to their temperature dependence. For this approach, only data from the peak vegetation period, i.e. June and July, should be used. Ideally, also the window length for the fit of eq. 7 and determination of parameters Pmax and k should be reduced.

Technical Corrections

Line 66: Use same units as in line 64, i.e. km^2.

Line 72: Is there Permafrost at all at this site, i.e. discontinuous Permafrost? Please clarify.

Line 302: Replace "somehow" with "to some extend".

Line 367: "GPP normalized by its model" is ambiguous. Use "(NEE - Rmod)/GPPmod".

References

Foken T, Wichura B (1996): Tools for quality assessment of surface-based flux measurements. Agricultural and Forest Meteorology, 78, pp. 83-105. DOI:10.1016/0168-1923(95)02248-1.

Gažovič et al. (2013): Hydrology-driven ecosystem respiration determines the carbon

balance of a boreal peatland. Science of the Total Environment, 463-464, pp. 675–682.

Mahecha et al. (2010): Global Convergence in the Temperature Sensitivity of Respiration at Ecosystem Level. Science, Vol. 329, Issue 5993, pp. 838-840. DOI: 10.1126/science.1189587.

---

## Referee Comment (RC2) · Anonymous Referee #2 · 3 Apr 2017

The manuscript presents net ecosystem and energy fluxes from a West Siberian bog measured by eddy covariance technique. This manuscript provides important data from a remote and large but also understudied region which is characterized by high coverage of peatlands. Unfortunately, the data presented here is just for 4 summer months but as it was measured at an established field station more data will surely follow in the future. The topic of the manuscript is well within the scope of the journal and the manuscript meets well the basic scientific quality. However, extensive revisions are necessary before publication of manuscript. Main points of critique 1. It would be important to include some information on vegetation development over the investigated period to better understand the dynamic of the $CO_2$ fluxes. 2. The gap filling procedures should be described in more detail. It should be clarified in the text and in the figures if modelled or measured data has been used and also discussed how the gap

filling method might influence the modelled data. If gap filling was not applied it should be discussed which influence it would have on the results. 3. The discussion is weak, maybe that was the reason why results and discussion were merged in one chapter. This chapter includes mainly comparison to other studies and less discussion of the influencing factors which determine the dynamics of $CO_2$ and heat fluxes. Specific comments: L 38: Please include some examples for measurements in Europe and in Siberia. L 55 Please change to flux tower data L 114 I usually include the measurement section into the methods section L128-130 You do not present winter fluxes here, so you can skip that paragraph or when does the winter start? L205 Why do you use the solar elevation angle and not the widely used PAR<10 $\mu$mol/m2s threshold to define the night-time? L 206 Did you try to use other soil temperatures than at 5 cm depth to model the respiration? Please include R2 to the Figure 3. L234-236 You use just the soil temperature at 5 cm depth for modelling, please include this information to the text and skip the information on soil temperatures at other depths. L 310-311 So the range of the values in the Fig 8a show just values from +3 to -9 $\mu$mol/m2s. L 312 The vegetation might play an important role as well. L 318 Please include the gapfilling methods for the NEE fluxes. L 329 What was the range of daily fluxes in Mukhrino? L 351 It might be interesting to include a figure with a typical diurnal course before and during the passage of the weather front.
* * *

---

## Referee Comment (RC3) · Anonymous Referee #3 · 5 Apr 2017

The paper by Alekseychik et al. presents the first data collected by an eddy covariance tower near the Mukhrino field station in West-Siberia. Since there are very few eddy covariance towers in this part of the world, I'm confident that the data collected will be of great interest to a wide audience, including modelers that wish to test their simulations. However, the station has not yet been operational for a long time and this study therefore only presents the first four months of data. Unfortunately, this also means that little new scientific knowledge on the processes governing carbon exchange is presented in this paper apart from adding an extra data point on the map (although data from understudied areas is valuable in itself, obviously).

As the first paper from a new field station, it is important that the description of the data collection is complete and accurate, since it will be the reference paper for future studies from this location. But in order to achieve that, several improvements need to

be implemented. Many of such remarks were already made by the other two referees, and I will therefore not delve to long on the areas where our reviews overlap.

First of all, it is not clear why the gapfilling and partitioning of the data has not been done by more common methods provided by the Fluxnet community. I suggest to do the partitioning and gapfilling of NEE according to Reichstein et al. (2005). The scripts to do so are freely available on the Fluxnet website. Following common Fluxnet methodology is important to include this new station as a valuable point of reference, and it would be helpful to point out where calculations are similar to previous studies, and where they diverge. Reference to Aubinet al. (2001) or the later book from Aubinet et al. (2012) are useful in that regard.

Also, as pointed out by the other referees, the paper does not present vegetation data and assumes too much about the phenology of the vegetation. If this data is unavailable, that would be a pity, since it would help a lot more to explain the data. Care needs to be taken to acknowledge that data gap, if it exists, and to not over-interpret the data.

Figure 1d shows that the area within the footprint has quite a bit of variance. The heat fluxes are integrated over this footprint, while soil temperature and net radiation measurements are taken in one point. An energy balance closure of 90% is then very high, given the fact that the different energy fluxes are not measured on the same area. The one place where some wiggle room remains is in calculation of the heat flux, which is highly dependent on the volumetic heat capacity of the soil in equation 4. Yet, soil properties are not mentioned in the paper and simply assumed to be 95% water and 5% peat, according to a reference from 1999. How realistic is this assumption and would your energy balance be worse if it was 80% water and 20% peat, to name just a number? Some uncertainty assessment of the assumptions behind the calculated soil heat flux would be preferable and show how this relates to the energy balance closure.

Some detailed comments:

Page 2, line 61: It would be good to include a more precise location of the tower, rather than these rounded coordinates, for future reference and model work.

Page 4, line 115: please mention the exact dates here also, and not only later in the document.

Page 6, line 169: is $G$ calculated from the soil temperature measurements at 2, 5, 10, 20 and 50 cm depth? Please specify.

Page 6, line 179: how well would this equation work for this site? Seems to me that volumetric water content would vary a lot between ridges and hollows.

Page 7, line 206: as mentioned by the others, why not simply look at measured PAR as a threshold? Was the sensor shaded by trees?

Page 7, line 210: fitting this equation on all data at once leads to a very uncertain fit, as is clear from Figure 3a, due to temporal variation in the base parameters. The method by Reichstein et al. (2005) therefore shifts short optimization windows throughout the season. Something similar should've been applied here, since $Q_{10}$ is probably not stable and depends on changes in e.g. soil moisture and substrate availability.

Page 7, line 216: same as previous remark. Why fit this to the entire dataset when the phenology of the plants, and therefore base parameters, is changing throughout the summer? There are better partitioning methods out there.

Page 8, line 231: 'dramatic' is a subjective term. Perhaps this is normal in this area?

Page 10, line 259: 'probably'? how would you know if you haven't measured this? Isn't the lower amount of incoming PAR the reason that $R_n$ is also lower?

Page 10, line 260: incoming solar radiation is logically lower in August, since it's further removed from midsummer. So this would also happen if there was no difference in cloud cover.

Page 14, line 345-347: This sentence is unclear. Are you talking about this in general

terms or are you referring to this site?

Page 15, line 364: The landscape in your footprint doesn't look homogenous at all, with all the variation between ridges and hollows. It's just that this variation is similar within different areas of your footprint, but that's not the same as homogeneity.

Figure 10: how was does normalizing done? Please explain.

Page 16, line 384: Are these observations really in these IPCC reports? Surely, there's a peat synthesis product out there that can be cited instead.

Page 16, line 388: You cannot say 'apparently' since you are not reporting the course of vascular plant leaf area development.

References

Aubinet, M., Grelle, A., Ibrom, A., Rannik, U., Moncrieff, J., Foken, T., Kowalski, A. S., Martin, P. H., Berbigier, P., Bernhofer, C., Clement, R., Elbers, J., Granier, A., Grünwald, T., Morgenstern, K., Pilegaard, K., Rebmann, C., Snijders, W., Valentini, R. and Vesala, T.: Estimates of the annual net carbon and water exchange of forests: The EUROFLUX methodology, Adv. Ecol. Res., 30, 113–175, 2000.

Aubinet, M., Vesala, T. and Papale, D.: Eddy Covariance, edited by M. Aubinet, T. Vesala, and D. Papale, Springer Netherlands, Dordrecht. 2012.

Reichstein, M., Falge, E., Baldocchi, D. D., Papale, D., Aubinet, M., Berbigier, P., Bernhofer, C., Buchmann, N., Gilmanov, T., Granier, A., Grünwald, T., Havrankova, K., Ilvesniemi, H., Janous, D., Knohl, A., Laurila, T., Lohila, A., Loustau, D., Matteucci, G., Meyers, T., Miglietta, F., Ourcival, J. M., Pumpanen, J., Rambal, S., Rotenberg, E., Sanz, M., Tenhunen, J., Seufert, G., Vaccari, F., Vesala, T., Yakir, D. and Valentini, R.: On the separation of net ecosystem exchange into assimilation and ecosystem respiration: review and improved algorithm, Global Change Biol., 11(9), 1424–1439, doi:10.1111/j.1365-2486.2005.001002.x, 2005.

---

## Author Comment (AC1) · 23 May 2017

**General Comments**

The manuscript presents energy and CO2 flux data from the West Siberian Taiga. This is valuable data, as the West Siberian Lowland is a vast understudied region. The presented 4-month data set is the first data of what is to become a permanent flux measurement site. Thus is can provide a base line for comparison with other sites and with data that will be collected at the same site in the years ahead.
Generally, the style of the manuscript and the presentation of data is adequate. However, the data analyses lag behind the state of the art and the discussion of the results is often weak. Extensive revisions are necessary before publication of the manuscript.
We express our gratitude for the time invested in deep analysis of our manuscript and for the many useful comments that have resulted from it.

**Main Critique**

(1) No information on the gap-filling of energy fluxes is given. Was gap-filling not performed for H and LE? Monthly means of these values could be seriously biased if they are calculated based on non-gap-filled time series. Gap-filling should be performed in order to derive sound estimates of mean or cumulative fluxes, and the methods used should be clearly presented in the methods section.
You are correct, the original non-gapfilled energy fluxes were presented in the original MS draft, as we considered that the data series were complete enough (at least in May-July) for the means not to be biased. However, we understand the concern and have done gapfilling of the energy fluxes. Soil heat flux is calculated from gapfilled soil temperature and water level data, therefore, it's gap-free. The other fluxes are gapfilled according to the accepted routines, e.g. Falge et al. (2001). The more in-depth explanation can be found in the revised manuscript (section 2.5). However, the changes after gapfilling, in terms of average or cumulative values (including Bowen ratio) were small.

(2) How have the authors addressed the heterogeneity of soil and hydrological properties and hence ground heat flux (G)? Was soil temperature measured and G calculated for only one microform, hummock or hollow? As G could be expected to vary strongly between hummocks and hollows, a weighted average (based on surface area fractions) of G calculated for both microforms should be used. If G is available for only one microform, an estimate of the error induced by this approach should be added (which could also serve as a justification for this approach).
The originally presented soil heat flus was calculated for hollows, as they are the dominating microform within the EC footprint. To account for this comment in the revision, we calculate G as an area-weighted average for hummocks and hollows (using two replicates of temperature profile in each microform). At the same time, the hydrological differences are tackled by using the ridge and hollow water level measurements in the calculation of the corresponding heat fluxes.

(3) In my opinion, an instationarity test (e.g. Foken and Wichura, 1996) is start of the art and should be applied.

We have applied the instationarity test in the new version of the manuscript and it did not introduce a statistically significant change in the observed fluxes, because it mainly remove very small fluxes (those close to zero). For instance, only a fraction of the negative nocturnal CO2 flux values were removed by this filter. However, it appears that the new method of Re and GPP model parameter estimation is robust enough so that non-stationarity filtering can be implemented without significant increase in parameter uncertainty. This is done in the revision with a non-stationarity threshold of 1.

(4) No information on the seasonal vegetation development is given in the manuscript. Even if assessments/measurements of GAI or LAI are not available, a general description of the vegetation development is indispensable in order to put the observed flux data in context to the annual cycle of fluxes and drivers.

1)_ Unfortunately, no LAI or snow monitoring has been done during the study period. Therefore, only approximate qualitative estimation of the two parameters can be offered.

Additionally, as the measurements started directly after the end of snow melt, information on the snow/soil conditions immediately before the beginning of the measurement period should be added, if possible (snow height, snow water equivalent, beginning of snow melt, depth of frozen peat layer, beginning/end of peat thaw). This could be very helpful to understand the temporal development of fluxes at the beginning of the growing season.

2) The snowmelt is shown by the steep downward trend in PAR albedo, indicating the presence of patches of snow until about 3 May. However, all profiles, except one in the hollow, indicate freezing at -5 cm until 3[rd] May, and until about 6[th] May at -20 cm. Therefore, the snowmelt and peat thaw proceeded only over the first few days of the study. The quantities such as snow pack depth or snow water equivalent are, unfortunately, currently unavailable.

(5) The partitioning of measured net ecosystem exchange (NEE), particularly the modelling of ecosystem respiration (Re) appears to be not sound. In Detail: (a) Why are there significant negative fluxes in the Re vs. peat temperature (Tp) plot (Fig. 3a)?

After careful QA/QC I would ideally expect to see only few and small negative night time CO2 fluxes (predominantly at lower temperatures). Maybe, the application of an instationarity test could help removing these conspicuous data points? (b) The fit of eq. 6 to the Re vs. Tp data set (Fig. 3a) seems to have a low R^2. I'd like to see the R^2 and p values for this fit. (d) Generally, combining data from the period May-August in one fit of Re vs. Tp is likely to confound the seasonal development of Rref with its temperature dependence. This is reflected in the large temperature sensitivity (Q10 value) obtained by the fit. Fitting Re vs. Tp in a moving window of length 10: : :30 days would be more appropriate. If this would lead to unrealistic variations of the reference respiration (Rref) and Q10, the authors could constrain Q10 to a value around 1.5 (cf. Mahecha et al., 2010). This way, at least the variation of Rref could be assessed, which could give valuable insights into the seasonal vegetation development.

We faced challenges with energy supply at the Mukhrino station. With many cloudy and low-wind periods, frequent blackouts occurred, especially in the nighttime periods. Eventually, this led to the nocturnal data being scarce as it is. The existing nighttime data were only sufficient to construct one general fit of Re vs. Tp (Fig. 3a). Therefore, unfortunately, recalculation of both Re model parameters (Re_ref and $Q_{10}$) in a moving window does not seem possible. However, a different modeling/partitioning method was used instead. It offers a tradeoff between robustness, precision and the ability to resolve the seasonal course of the parameters, incorporating the following steps:

The updated modeling/gapfilling approach
a) The complete NEE equation, NEE = Rref*Q10^((Tmoss-12)/10) - (Pmax*PAR)/(k+PAR)),
 is fit to the data at PAR<300 W/m2, in the region where exchange is dominated by respiration.
This fit yields Q10=1.99, 95% CI [1.42; 2.57]. This value of Q10 is fixed for the entire May-August
period. Regarding the Reviewer's remark on Q10, we would in return suggest that it mainly stems
from the choice of the driving temperature, and, as such, does not carry much biological meaning.
b) The Rref, Pmax and k parameters are evaluated in a 30-day wide moving time window.
c) The Rref, Pmax and k series are spline-interpolated to produce the 30-min series, after which the
models can be calculated at the original data resolution.
The overview of the gapfilling method is included in the revision (section 2.6).

The superior performance of the described method, compared with its older version, is revealed by
a smaller model-measured intercept (see e.g. the figure with the mean diurnal CO2 flux course).

Flux nonstationarity filtering.
Filtering the CO2 flux for high nonstationarity does remove a number of nighttime data in addition
to the u* filter; however, the data points so excluded are randomly distributed over the entire
nighttime Re range (i.e. both negative and positive values are affected). The negative values in
nocturnal Fco2 likely result from high random uncertainty, i.e. they are counterweighted by
similarly high positive values. In the revised version of the manuscript, the FST filter is used with
the threshold of 1.

To sum it up, we believe that the mean flux-temperature regression, in both its old and new
versions, is realistic. This is supported by the fact that the general Re model of Mukhrino does look
similar to those found in the other sites. For example, in a similar Siikaneva-2 site (Southern
Finnish bog), an ensemble of data from 4 growing seasons showed $Rref = 0.8$ µmol $m^{-2} s^{-1}$ and $Q_{10} =$
3.5.

**Specific Comments**

Line 123: On which micro-form was soil temperature measured (hummock/hollow)?
See (2) in Section "Main Critique" above.
The presented soil temperature and the derived heat flux were measured at a hollow microform.
However, in the revision we use a total of four profiles, in hollow and ridge. The area-weighted
average ridge-hollow temperature time series are shown in Fig. 4a.

Line 169: For which micro-form was ground heat flux calculated (hummock/hollow)?
See (2) in Section "Main Critique" above.
In the original MS version, at the hollow microform. However, the difference between the updated
and old G is not great, in correspondence with the dominance of hollows.

Lines 193-194: Why was only CO2 night time data of August excluded from analysis?
It is hard to imagine that only CO2 fluxes are compromised by technical problems of
the gas analyzer but not LE fluxes.
The source of the problem affecting the August nocturnal CO2 flux is not known. The Rref
parameter increases notably in August, and it is difficult to say if this is a natural dynamic or a
technical problem. The objective quality criteria do not remove those data. However, LE seems not

to be affected, as its August nighttime values are close to zero as in the previous months. In any case, we decided to keep the August data, but be tentative in its interpretation.

Lines 205-206: Why was night-time defined as periods with a solar elevation angle below 5_ and not by a PAR threshold (e.g. PAR < 20 _mol/m2/s)? Using a local PAR threshold may allow additional data points to be included into the night time data set (e.g. during cloudy conditions), which could improve the data coverage and hence the modelling of Re.
The night definition was updated as proposed, as the periods with PAR<10 umol m-2 s-1.

Line 206: Have you tried to use peat temperature from other depths or air temperature for the modelling of Re? Information on the performance of the model with other temperatures could give valuable insights into the source of respired CO2.
Yes, we did try to use the other temperatures as drivers of respiration. The hummock temperature measurement had been previously used for that purpose. This is consistent with much higher density of vegetation and low water level in hummocks, which probably makes them major contributors to ecosystem respiration despite representing a smaller area fraction than hollows. However, in consistency with the revised soil heat flux, we are now using the area-weighted soil surface temperature also in Re modeling.

Lines 219-221: In which time steps was the 30-day window moved? Please add this information. Further, the time series of the fit parameters Pmax and k (or the often used alpha = Pmax/k) should be presented. This could deliver valuable information on the seasonal development of the vegetation and could be compared to other studies.
The time window was moved in 1 day steps. We will present the Pmax, k and alpha parameter timeseries in a new figure or subplot. Please also see the requested data in Fig. R1 below.

[Figure]

Figure R1. The timeseries of the CO2 flux model parameters. The dots are the daily values estimated in a moving window 30 days wide; the solid lines are the spline interpolants, and the shaded area is the 95% confidence interval (calculated in each time window). This figure will be included in the revised manuscript.

Lines 234-236: Soil temperature at depths 20 cm and 50 cm is discussed here, but this data is neither displayed in Fig. 4a nor used in the analyses. I suggest to either add this data in an additional subplot of Fig. 4 or concentrate in the text on the data already displayed in Fig. 4, i.e. Tp at 5 cm depth.

This mismatch between the text and the plots in Fig.4 is confusing indeed. We will add the 20 and 50 cm temperatures in the subplot (a) of Fig.4.

Lines 258-259: The statement ": : :later on during the summer the water level decreases: : :" contradicts what is shown in Fig. 4e, and is stated in lines 344-345, "The regular and ample precipitation helped sustain water level at a nearly constant level: : :". Hence, the authors' assumption that albedo is reduced due to drying of the vegetation is ill-conceived. Still, it could be checked by simply calculating an albedo from incoming and reflected PAR.

[Figure]

Figure R2. PAR albedo for the hollow and hummock microforms calculated as diurnal medians of the midday (10AM-16PM) periods. The grey dots are the original 30-min albedo averages. WTD (black) and precipitation (purple) proxies are also shown for reference.

Our original expectation was to see an increase in albedo during the dry spells (lines 258-259), which is commonly observed in other peatlands worldwide. However, the year 2015 being unusually wet, the water table did not follow its typical downward trend, nor did dry spells last long. In fact, there were at least seven periods of WTD drawdown and subsequent recovery during heavy rainfalls. On average, WTD had maybe remained constant throughout the growing season. The time-series of albedo are shown in Fig.R2 above. Albedo was rather stable at about 0.06 in hollow and 0.04 in hummock, although small variation correlated with WTD and precipitation can be seen. Similar peatland PAR albedo values were found in other studies (e.g. 5.5% in Frolking et al. 1998). In general, the correlation with rainfall seems to be higher than that with WTD, which is

consistent with the expectation that surface wetness is a stronger controller than WTD. WTD may be decoupled from surface wetness, which is especially probable in hummock, meaning that WTD is probably an inferior predictor of surface wetness and, hence, albedo. In this sequence of frequent rewetting events and drying periods, the surface wetness and albedo shows simultaneous peaks, which is illustrated well by the hummock measurements (Fig. R2).
Of course, the phenology (course of LAI, etc.) in sedge in other vascular species must have affected albedo in a way that is difficult to estimate for the lack of observations. Qualitatively, the dark-colored living vascular plant biomass should lower the ecosystem albedo around the peak of the growing season.
Also, note the steep albedo plunge in early May, indicative of the final snowmelt stage.

Line 302: The spatial heterogeneity does not seem to serve as a good explanation for the low value of the energy balance closure in May, as the surface heterogeneity does not change during the course of the measurement period. Or does it change? How?
The change in the area of open water pools is mentioned as one of the possible sources of heterogeneity in May (line 302), which, in turn, may affect the ground heat flux (line 303). This was the month when the snowmelt ended, which is typically accompanied by scattered open water pools. Their locations were not recorded. We do not know how the surface energy balance measurements could have been affected by those spring conditions. The ground heat flux in hollow was very variable during May, reaching the highest levels for the whole growing season in mid-May with a subsequent reduction to the average annual levels (Fig. R3), suggesting some rapid changes in the thermal properties of the ground.

[Figure]

Figure R3. Timeseries of the ground heat flux in hollow, precipitation and WTD. The units are arbitrary.

Line 310 and Fig. 8a caption: The data displayed is surely modelled NEE and not measured NEE?
We apologize for the inconsistency - this is actually gapfilled NEE, i.e. the quality-controlled and u*-filtered original NEE record with the gaps filled by the NEE model.

Lines 311-312: Could the lower amplitude of NEE in May also be due to a not fully developed foliage of the vegetation? Snow melt had only ended a few days before and below-zero temperatures still seem to occur during May. Time series of the parameters Pmax, k, and Rref could help to explain the variations in observed NEE.

You are absolutely right. The ground vascular vegetation (shrubs, sedges) would have only started to recover from the winter and grow their leaf area. While Rref cannot be resolved on a finer timescale for the reasons discussed above. Exactly as suggested, Pmax shows a steep upward trend between May and June, the period of green biomass accumulation. The time-series of the model parameters are now shown in a Figure 3 in the revised manuscript.

Lines 213-314: I see a systematic difference of measured and modelled NEE in Fig. 9. In the afternoon hours of July, measured NEE uptake is smaller than modelled NEE uptake. Hence, either Re is underestimated or GPP is overestimated. What could be the reason for this? Furthermore, why is August night time CO2 flux data displayed if it should have been excluded from analysis due to technical problems (line 194)? In fact, this data does not look completely unrealistic to me. Gažovi˘c et al. (2013) has observed the highest Re during August, while GPP peaked in July. The discrepancies between modelled and measured fluxes could be caused by the fact that Re is poorly modelled by the approach chosen by the authors.

The general observation relating to the new modeling results is that this mismatch is mainly gone. It seems that, just as you have suggested, the model-measured NEE mismatch had been caused by the suboptimal Re modelling method and/or the choice of the driving temperature. In any case, the updated version of the NEE model is probably good enough for the gapfilling purposes.
We also agree (as in the response to an earlier comment) that the August respiration data might be correct. The hypothesis about the technical problems in August has not been confirmed after a cross-check; it was established during revision that no objective filtering or quality control steps could remove the August data. It is also encouraging to hear that some studies, including Gažovic et al. (2013), found similar seasonal trends, thank you for pointing this out. However, data coverage in August is still very low, and the results from that month should be treated with caution.

Lines 352-353 and Fig. 10: Combining all data from the period May-August potentially confounds the seasonal development of Pmax and k, and hence GPPmod, with a possible short term variation of these parameters due to their temperature dependence.
For this approach, only data from the peak vegetation period, i.e. June and July, should be used. Ideally, also the window length for the fit of eq. 7 and determination of parameters Pmax and k should be reduced.

We understand the logic behind this comment. However, as the GPP model used for normalization was calculated from the seasonally changing Pmax and k series, it thus implicitly includes LAI development and other low-frequency seasonal factors, although short-term variability may be lost. We have experimented with different time windows of Pmax and k, and found that at lengths shorter than 1 month, the random variability increasingly dominated the real signal. This was due to the gaps in the original 30-min data, which can only be circumvented by using a sufficiently wide time window. Nevertheless, since May represents the spring recovery period, and August is not covered with data well, using only June and July would improve this analysis. However, for June-July, the picture remains about the same (Fig. R4). Reducing the window length to under 1 month is problematic, due to the scarcity of the nighttime data, but we believe that the current length allows evaluating the seasonal change, at the least.
The observed NEE normalized with the NEE model vs. air temperature for June-July 2015 can now be seen in Fig. 10b of the revised manuscript.

**Technical Corrections**

Line 66: Use same units as in line 64, i.e. kmˆ2.
Done

Line 72: Is there Permafrost at all at this site, i.e. discontinuous Permafrost? Please clarify.
There is no permafrost of any type in Mukhrino or anywhere else in the region; this is clarified in the revision.

Line 302: Replace "somehow" with "to some extend".
Done

Line 367: "GPP normalized by its model" is ambiguous. Use "(NEE - Rmod)/GPPmod".
Done

**References**

Foken T, Wichura B (1996): Tools for quality assessment of surface-based flux measurements. Agricultural and Forest Meteorology, 78, pp. 83-105. DOI:10.1016/0168-1923(95)02248-1.

Gažovič et al. (2013): Hydrology-driven ecosystem respiration determines the carbon balance of a boreal peatland. Science of the Total Environment, 463-464, pp. 675–682. Mahecha et al. (2010): Global Convergence in the Temperature Sensitivity of Respiration at Ecosystem Level. Science, Vol. 329, Issue 5993, pp. 838-840. DOI: 10.1126/science.1189587.

Frolking, S.E., Bubier, J.L., Moore, T.R., Ball, T., Bellisario, L.M., Bhardwaj, A., Carroll, P., Crill, P.M., Lafleur, P.M., McCaughey, J.H., Roulet, N.T., Suyker, A.E., Verma, S.B., Waddington, J.M. and Whiting, G.J. (1998). Relationship between ecosystem productivity and photosynthetically active radiation for northern peatlands. Global Biogeochemical Cycles 12: doi: 10.1029/97GB03367.

Falge, E., Baldocchi, D., Olson, R., Anthoni, P., Aubinet, M., Bernhofer, C., Burba, G., Ceulemans, R., Clement, R., Dolman, H., Granier, A., Gross, P., Grunwald, T., Hollinger, D., Jensen, N.O., Katul, G., Keronen, P., Kowalski, A., Lai, C.T., Law, B.E., Meyers, T., Moncrieff, J., Moors, E., Munger, J.W., Pilegaard, K., Rannik, U., Rebmann, C., Suyker, A., Tenhunen, J., Tu, K., Verma, S., Vesala, T., Wilson, K., Wofsy, S., 2001. Gap filling strategies for defensible annual sums of net ecosystem exchange. Agric. Forest Meteorol. 107, 43–69.

---

## Author Comment (AC2) · 23 May 2017

The manuscript presents net ecosystem and energy fluxes from a West Siberian bog measured by eddy covariance technique. This manuscript provides important data from a remote and large but also understudied region which is characterized by high coverage of peatlands. Unfortunately, the data presented here is just for 4 summer months but as it was measured at an established field station more data will surely follow in the future. The topic of the manuscript is well within the scope of the journal
and the manuscript meets well the basic scientific quality. However, extensive revisions are necessary before publication of manuscript.
Dear reviewer, we express our gratitude for your detailed analysis of the current manuscript. We will try to address all the comments.

**Main points of critique**

1. It would be important to include some information on vegetation development over the investigated period to better understand the dynamic of the CO2 fluxes.
Unfortunately, the vegetation parameters (such as LAI, biomass or phenology) were not directly assessed during the course of the measurements. Therefore , we can only offer qualitative estimates.

2. The gap filling procedures should be described in more detail. It should be clarified in the text and in the figures if modelled or measured data has been used and also discussed how the gap filling method might influence the modelled data. If gap filling was not applied it should be discussed which influence it would have on the results.
This is correct, the CO2 flux data series have been gapfilled already in the original MS, and both the gapfilled and non-gapfilled series were displayed throughout the manuscript, depending on the aims of the individual sections. We will specify this more clearly in the text and explain the gapfilling procedure in more detail. The modeling/gapfilling method has also been revised; please see the detailed explanation in the answer to a comment by Reviewer 1. Please see the revised MS sections 2.5 and 2.6 for the description of the updated gapfilling procedures for the energy and CO2 fluxes.

3. The discussion is weak, maybe that was the reason why results and discussion were merged in one chapter. This chapter includes mainly comparison to other studies and less discussion of the influencing factors which determine the dynamics of CO2 and heat fluxes.
We understand this comment and will try to improve the discussion. We will revise the discussion of the environmental driver effects as much as it is possible with the available dataset, as much as the available data permits.

**Specific comments:**

L 38: Please include some examples for measurements in Europe and in Siberia.
We will mention the relevant examples of bog measurements in the region. However, very few other peatland studies have been conducted in the region – this not only including West-Siberian middle taiga, but in Siberia as a whole. The number of relevant studies becomes even more limited

if one considers only raised bog ecosystems. The revised MS lines 39-44 now list the comparable sites having eddy-covariance setups.

L 55 Please change to flux tower data
Thank you for noticing this slip of the pen, this will be corrected.

L 114 I usually include the measurement section into the methods section
It will maybe make more sense indeed to combine the 'Materials' and 'Methods' sections; this is done in the revised version.

L128-130 You do not present winter fluxes here, so you can skip that paragraph or when does the winter start?
This is in fact the first description of the measurement site and its infrastructures that has not appeared in any prior English language publications. We therefore tried to include all the information that could potentially be of interest. In fact, in 2015 the snowmelt ended in the very beginning of May, as suggested by the PAR albedo timeseries (Fig. R2 in the response to Reviewer 1).

L205 Why do you use the solar elevation angle and not the widely used PAR<10 _mol/m2s threshold to define the night-time?
The solar elevation angle criterion has also been widely used in our community. A possible issue with using the PAR threshold is that PAR is measured at about 2m height, while the eddy-covariance sensors are installed at 4m height. Furthermore, Fig. A1 suggests that there is a great degree of overlap between the two night definitions, as far as the mean $CO_2$ flux is concerned. As one can see, on average, the CO2 flux transits zero somewhere near +5 degrees threshold that we have defined. At the same time, when the PAR threshold of 10 umol m-2 s-1 is used, an equivalent number of negative CO2 data are marked as "night". So, we suggest that the two methods of night period definition produce approximately equivalent results. Nevertheless, we are using the PAR definition in the revised manuscript.

[Figure]

**Fig.A1**. EC $CO_2$ flux versus solar elevation angle.

L 206 Did you try to use other soil temperatures than at 5 cm depth to model the respiration? Please include R2 to the Figure 3.
Yes, we did. It was difficult to find any improvement when the other temperature measurements were tried. In the revision, we are using the area-weighted average of the ridge and hollow -5cm temperatures, so that to account for the spatial variation.
The original Figure 3 proved to be confusing and gave a wrong idea that the displayed "general" fits were used to model Re and GPP for the whole period, therefore, it was replaced with the parameter timeseries figure. We did not use the general regression of peat temperature to Re anymore, as described in the description of the revised gapfilling method.
In any case, the requested R2 of the Tp-Re fit in former Figure 3a was 0.55.

L234-236 You use just the soil temperature at 5 cm depth for modelling, please include this information to the text and skip the information on soil temperatures at other depths.
We would maybe argue that the 20 and 50 cm depths be kept, once again for the reason that it has not been published anywhere, while it might interest some as background. The 20 and 50cm depths are added to Fig. 4a in the revision.

L 310-311 So the range of the values in the Fig 8a show just values from +3 to -9 mol/m2s.

This is correct. We did zoom into the main data cluster so that to make the seasonal dynamics clearer. The extreme 30min averages in fact do not need to be mentioned, as they represent the typical random variability of the eddy-covariance fluxes rather than the typical exchange, which is bounded by about +5 and -10 umol/m2s.

L 312 The vegetation might play an important role as well.
You are absolutely right, we should mention this biological driver along with the abiotic ones, temperature and PAR. However, the LAI or other vegetation parameters have not been directly measured at the site.

L 318 Please include the gapfilling methods for the NEE fluxes.
We agree on the importance of this information; we hope that the revised section 2.6 provides sufficient detail on the gapfilling method.

L 329 What was the range of daily fluxes in Mukhrino?
For June-August, the average daily cumulative NEE was -1.8 g C m-2, so, well within the values observed elsewhere. This estimate is included in the revised text for a clearer comparison with the literature.

L 351 It might be interesting to include a figure with a typical diurnal course before and during the passage of the weather front.
It should be possible to add show a close-up of the daily course as an extra panel to Fig.8. This addition is made in the revised MS. The third front passage period having the least proportion of gaps is displayed as a case study.

---

## Author Comment (AC3) · 23 May 2017

The paper by Alekseychik et al. presents the first data collected by an eddy covariance tower near the Mukhrino field station in West-Siberia. Since there are very few eddy covariance towers in this part of the world, I'm confident that the data collected will be of great interest to a wide audience, including modelers that wish to test their simulations. However, the station has not yet been operational for a long time and this study therefore only presents the first four months of data. Unfortunately, this also means that little new scientific knowledge on the processes governing carbon exchange is presented in this paper apart from adding an extra data point on the map (although data from understudied areas is valuable in itself, obviously).

We thank you for you appreciation our work and acknowledging the novelty of the study. We certainly are planning to continue the measurements in as continuous manner as it will be possible.

As the first paper from a new field station, it is important that the description of the data collection is complete and accurate, since it will be the reference paper for future studies from this location. But in order to achieve that, several improvements need to be implemented. Many of such remarks were already made by the other two referees, and I will therefore not delve to long on the areas where our reviews overlap. First of all, it is not clear why the gapfilling and partitioning of the data has not been done by more common methods provided by the Fluxnet community. I suggest to do the partitioning and gapfilling of NEE according to Reichstein et al. (2005). The scripts to do so are freely available on the Fluxnet website. Following common Fluxnet methodology is important to include this new station as a valuable point of reference, and it would be helpful to point out where calculations are similar to previous studies, and where they diverge. Reference to Aubinet al. (2001) or the later book from Aubinet et al. (2012) are useful in that regard.

We understand the concern. The modeling/partitioning method has been updated in the revised manuscript, and the explanation can be found in the response to a comment by Reviewer 1. The new method improves the models by accounting for the seasonal change in the respiration parameter Rref.

However, given such a challenging dataset, the strict application of the Fluxnet methods seems problematic, as those methods function more optimally with more complete datasets. This was acknowledged by Reichstein et al. 2005:

"To sum up, the algorithm introduced here was able to find a short-term temperature response of Reco at all studied sites and is a significant step forward towards less biased estimates of Reco and GEP. Nevertheless, important limitations should be noted: It is not guaranteed to work at all sites since whether one can find a reliable short-term relationship between Reco and temperature depends on the noisiness of the eddy data and the range of temperatures encompassed during the short period. At sites with very stable temperatures and noisy eddy covariance data, it might be possible that within a year no short period can be found where a temperature–Reco relationship can be established at all. Seasonal changes in the temperature sensitivity that have been hypothesized are hard to detect from eddy covariance data, since in many cases not enough shortterm periods with a good correlation between temperature and Reco were found to make up a seasonality."

We have to admit that our data is noisy in exactly this sense, and, in addition, it is full of gaps, especially during the nights.

Nevertheless, we believe that our approach to Re and GPP modeling works towards the same ends as that of Fluxnet, while being at the same time optimized for our dataset.

Also, as pointed out by the other referees, the paper does not present vegetation data and assumes too much about the phenology of the vegetation. If this data is unavailable, that would be a pity, since it would help a lot more to explain the data. Care needs to be taken to acknowledge that data gap, if it exists, and to not over-interpret the data.
Unfortunately, LAI has not yet been determined at the site.

Figure 1d shows that the area within the footprint has quite a bit of variance. The heat fluxes are integrated over this footprint, while soil temperature and net radiation measurements are taken in one point. An energy balance closure of 90% is then very high, given the fact that the different energy fluxes are not measured on the same area. The one place where some wiggle room remains is in calculation of the heat flux, which is highly dependent on the volumetric heat capacity of the soil in equation 4. Yet, soil properties are not mentioned in the paper and simply assumed to be 95% water and 5% peat, according to a reference from 1999. How realistic is this assumption and would your energy balance be worse if it was 80% water and 20% peat, to name just a number? Some uncertainty assessment of the assumptions behind the calculated soil heat flux would be preferable and show how this relates to the energy balance closure.
The 95% porosity in the top 50 cm of peat was originally adopted from Yurova et al. 2007, whose model was used to predict the profile of volumetric water content based on water level, as we lacked direct observations of water level. The Yurova et al. model was parameterized for the porosity values ranging from 92 to 98% between catotelm and acrotelm, so we assumed that 95% would represent the mean conditions well enough. However, at the time we were preparing this manuscript, the new results on the physical properties of peat at Mukhrino were not yet available. Szajdak et al. (2016) summarized their measurements in six representative micro-landscapes around the site and found an average porosity in the top 50 cm of peat to be 93%. Therefore, our earlier assumption of 95% porosity in surface peat layer was realistic. To be consistent with Szajdak et al. (2016), we will update the porosity value.
The other reviewers suggested to recalculate soil heat flux as an area-weighted average of the fluxes calculated for hummocks and hollows, which we do in the revised MS version. As Szajdak et al. (2016) do not present microform-specific results, we will be forced to use the same 93% porosity for both microform types.

Page 2, line 61: It would be good to include a more precise location of the tower, rather than these rounded coordinates, for future reference and model work.
Will be done. The precise coordinates of the EC tower are given on the line 126 in the revised MS.

Page 4, line 115: please mention the exact dates here also, and not only later in the document.
Done.

Page 6, line 169: is G calculated from the soil temperature measurements at 2, 5, 10, 20 and 50 cm depth? Please specify.
It was calculated from the temperature measurements at 5, 10, 20 and 50 cm depths. This will be specified. The 2 cm level was omitted because there was no certainty that it shows the peat (moss) but not air temperature, i.e. that the sensor was constantly in a good contact with the vegetative parts or soil.

Page 6, line 179: how well would this equation work for this site? Seems to me that volumetric water content would vary a lot between ridges and hollows.
The Yurova et al. 2007 model was constructed for a Swedish fen, which closely resembles the lawn microsites of Mukhrino. Lawns-hollows being the dominating microform in the areal sense, the equation probably describes the average water content over most of the area with satisfactory precision. In the revision, the hollow and ridge WTD data are used separately to model water content profile for these microsite types.

Page 7, line 206: as mentioned by the others, why not simply look at measured PAR as a threshold? Was the sensor shaded by trees?
The $CO_2$ flux distributions at the two night definitions are more or less identical (Fig. R1). These data are u*-filtered. As with the of the open-path sensors in general, our nighttime flux is characterized with high random uncertainty. Both definitions yield realistic mean nocturnal $CO_2$ flux of about +1 µmol m-2 s-1, have similar shapes and kurtoses. However, the definition based on solar angle leads to more nocturnal periods than the PAR definition (610 versus 455).
In any case, we are using the PAR definition of nighttime periods in the revision, as the new $CO_2$ flux model is not as sensitive to the amount of nighttime data, as its previous version.

[Figure]

Fig. R1. Histograms of eddy-covariance $CO_2$ flux for the two alternative night definitions, based on PAR (left panel) and solar angle (right panel).

Page 7, line 210: fitting this equation on all data at once leads to a very uncertain fit, as is clear from Figure 3a, due to temporal variation in the base parameters. The method by Reichstein et al. (2005) therefore shifts short optimization windows throughout the season. Something similar should've been applied here, since Q10 is probably not stable and depends on changes in e.g. soil moisture and substrate availability.
As discussed in the response to the previous comment, the number of nocturnal data is very low, and this prompted us to use that simplified approach. Representation of the seasonal course in Q10 seems challenging for this dataset, however. However, we found a way to circumvent the problem by representing the seasonality in the Rref parameter. This is done in a way similar to the Reichstein et al. 2005, although not down to fine detail. The limitation of the data coverage has played a major role in the choice of the gapfilling method, however we hope that we reach similar targets as Reichstein et al. 2005.

Page 7, line 216: same as previous remark. Why fit this to the entire dataset when the phenology of the plants, and therefore base parameters, is changing throughout the summer? There are better partitioning methods out there.

We must have described the gapfilling scheme in a confusing way, apologies for this. In fact, the other reviewers were similarly puzzled about this point. Fig. 3 shows a fit to all data together just for demonstration. What actually was done, is the estimation of the GPP model parameters (Pmax and k) in a 1 month-wide moving time window moved in 1-day steps. Smoothened timeseries of Pmax and k were then obtained from these daily estimates. So, our approach does capture the vegetation phenology and other seasonal effects. Please see Fig. R1, which was also included in our response to Review 1. The updated $CO_2$ flux gapfilling method is described in the revised MS section 2.6.

Page 8, line 231: 'dramatic' is a subjective term. Perhaps this is normal in this area?

We wished to underline the dramatic nature of snowmelt in the year 2015, as it really was beyond the normal – early and rapid. In the revised manuscript, we rephrase it as "an unusually early and rapid snowmelt in April and the beginning of May".

Page 10, line 259: 'probably'? how would you know if you haven't measured this? Isn't the lower amount of incoming PAR the reason that Rn is also lower?

At the time of this response' writing, we can confirm this suggestion with data. We do observe a summer minimum in PAR albedo in early July (about 4%). After that, PAR albedo climbs to 6-7% by mid-August. However, while the short-term variations in PAR albedo are correlated with the heavy rain periods, the absence of strong seasonal trend in WTD does not allow to view it as a driver of PAR albedo in that year. Instead, the late summer increase in albedo must have been due to the senescence of vascular vegetation. Although albedo for global radiation was not measured, one could surmise that it followed similar trends.

Finally, Rn reduction solely due to change in albedo was meant here (line 259). Of course, this occurs in parallel with the downward trend in insolation after the summer solstice.

Page 10, line 260: incoming solar radiation is logically lower in August, since it's further removed from midsummer. So this would also happen if there was no difference in cloud cover.

We fully agree. As in the previous point, the individual effect of cloudiness was meant, as an addition to the astronomical component in solar radiation variation. We are sorry that this was not stated clearly, and it will be rephrased.

Page 14, line 345-347: This sentence is unclear. Are you talking about this in general terms or are you referring to this site?

If we understand this comment correctly, yes, we are discussing the effects of the fronts in terms of the *in situ* observations. The water table dynamics discussion in Price (2003) is of general relevance, and only used as a theoretical framework for the particular case of Mukhrino.

Page 15, line 364: The landscape in your footprint doesn't look homogenous at all, with all the variation between ridges and hollows. It's just that this variation is similar within different areas of your footprint, but that's not the same as homogeneity.

We agree with this note. This landscape is homogeneous on the length scales of 100 m and more, but it could be incorrect to call it homogeneous in a common sense (as in the Degerö which is devoid of any small or large scale features, e.g. Peichl et al. 2013).

Figure 10: how was does normalizing done? Please explain.

The observed flux was simply divided by the model. We will make this more specific in the text.

Page 16, line 384: Are these observations really in these IPCC reports? Surely, there's a peat synthesis product out there that can be cited instead.
It actually the "Supplement to the 2006 IPCC Guidelines for National Greenhouse Gas Inventories: Wetlands" we are referring to. IPCC themselves suggest referring to it as "IPCC 2013, 2014".

Quoting the IPCC website (http://www.ipcc-nggip.iges.or.jp/public/wetlands/):
"Please cite as: IPCC 2014, 2013 Supplement to the 2006 IPCC Guidelines for National Greenhouse Gas Inventories: Wetlands, Hiraishi, T., Krug, T., Tanabe, K., Srivastava, N., Baasansuren, J., Fukuda, M. and Troxler, T.G. (eds). Published: IPCC, Switzerland"

Page 16, line 388: You cannot say 'apparently' since you are not reporting the course of vascular plant leaf area development.
LAI was not measured and we can only surmise about its effects. However, the Pmax course (Fig. R2) should approximately correspond to the LAI seasonal curve. But we agree that using "possibly" or "likely" would suit this sentence more than "apparently".

**References**

Granberg, G., Grip, H., Löfvenius, M. O., Sundh, I., Svensson, B. H., and Nilsson, M.: A simple model for simulation of water content, soil frost, and soil temperatures in boreal mixed mires, Water Resour. Res., 35, 3771-3782, 10.1029/1999WR900216, 1999.

Peichl, M., Sagerfors, J., Lindroth, A., Buffam, I., Grelle, A., Klemedtsson, L., Laudon, H. and Nilsson, M.: Energy exchange and water budget partitioning in a boreal minerogenic mire, J. Geo.Res., 118, 1-13, 2013.

Szajdak L.W. et al. 2016. Physical, chemical and biochemical properties of Western Siberia. Environmental dynamics and global climate change. 2016. 7, pp. 13–25.

Yurova, A., Wolf, A., Sagerfors, J., and Nilsson, M.: Variations in net ecosystem exchange of carbon dioxide in a boreal mire: Modeling mechanisms linked to water table position, J. Geophys. Res.: Biogeosciences, 112, 10.1029/2006JG000342, 2007.

---

## Author Response (AR2)

Dear Editor, we are very glad to have received positive evaluation from you, and happy that the corrections were found satisfactory.

We have corrected the technical errors you have identified:

1) The correct PAR unit is of course $\mu mol\ m^{-2}\ s^{-1}$. The erroneous instances have been corrected accordingly (lines 243-244 in this file and Table 1).

2) The average climate data is coming from the meteorological station in Khanty-Mansiysk city, now mentioned in the text (line 83 in this file).

3) $NEE_{mod}$ meant in Fig.10 is the sum of Re and GPP models, as detailed in section 2.6. The explanation has been clarified (line 252).

[revised manuscript text omitted]